# Outcome-directed Reinforcement Learning by Uncertainty & Temporal Distance-Aware Curriculum Goal Generation

**Daesol Cho**[*]**, Seungjae Lee**[*]**, H. Jin Kim**
Seoul National University, Automation and Systems Research Institute (ASRI),
Artificial Intelligence Institute of Seoul National University (AIIS)
{dscho1234, ysz0301, hjinkim}@snu.ac.kr

## Abstract

Current reinforcement learning (RL) often suffers when solving a challenging exploration problem where the desired outcomes or high rewards are rarely observed. Even though curriculum RL, a framework that solves complex tasks by proposing a sequence of surrogate tasks, shows reasonable results, most of the previous works still have difficulty in proposing curriculum due to the absence of a mechanism for obtaining calibrated guidance to the desired outcome state without any prior domain knowledge. To alleviate it, we propose an uncertainty & temporal distance-aware curriculum goal generation method for the outcome-directed RL via solving a bipartite matching problem. It could not only provide precisely calibrated guidance of the curriculum to the desired outcome states but also bring much better sample efficiency and geometry-agnostic curriculum goal proposal capability compared to previous curriculum RL methods. We demonstrate that our algorithm significantly outperforms these prior methods in a variety of challenging navigation tasks and robotic manipulation tasks in a quantitative and qualitative way.[1]

## 1 Introduction

While reinforcement learning (RL) shows promising results in automated learning of behavioral skills, it is still not enough to solve a challenging uninformed search problem where the desired behavior and rewards are sparsely observed. Some techniques tackle this problem by utilizing the shaped reward (Hartikainen et al., 2019) or combining representation learning for efficient exploration (Ghosh et al., 2018). But, these not only become prohibitively time-consuming in terms of the required human efforts, but also require significant domain knowledge for shaping the reward or designing the task-specific representation learning objective. What if we could design the algorithm that automatically progresses toward the desired behavior without any domain knowledge and human efforts, while distilling the experiences into the general purpose policy?

An effective scheme for designing such an algorithm is one that learns on a tailored sequence of curriculum goals, allowing the agent to autonomously practice the intermediate tasks. However, a fundamental challenge is that proposing the curriculum goal to the agent is intimately connected to the efficient desired outcome-directed exploration and vice versa. If the curriculum generation is ineffective for recognizing *frontier parts* of the explored and feasible areas, an efficient exploration toward the desired outcome states cannot be performed. Even though some prior works propose to modify the curriculum distribution into a uniform one over the feasible state space (Pong et al., 2019; Klink et al., 2022) or generate a curriculum based on the level of difficulty (Florensa et al., 2018; Sukhbaatar et al., 2017), most of these methods show slow curriculum progress due to the process of skewing the curriculum distribution toward the uniform one rather than the frontier of the explored region or the properties that are susceptible to focusing on infeasible goals where the agent's capability stagnates in the intermediate level of difficulty.

---

[*]Equal contribution.
[1]Code is available : https://github.com/jayLEE0301/outpace_official

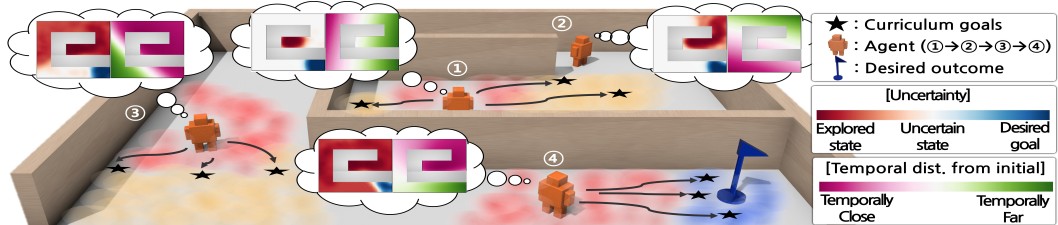

Figure 1: OUTPACE proposes uncertainty and temporal distance-aware curriculum goals to enable the agent to progress toward the desired outcome state automatically. Note that the temporal distance estimation is reliable within the explored region where we query the curriculum goals.

Conversely, without the efficient desired *outcome-directed* exploration, the curriculum proposal could be ineffective when recognizing the frontier parts in terms of progressing toward the desired outcomes because the curriculum goals, in general, are obtained from the agent's experiences through exploration. Even though some prior works propose the success-example-based approaches (Fu et al., 2018; Singh et al., 2019; Li et al., 2021), these are limited to achieving the only given example states, which means these cannot be generalized to the arbitrary goal-conditioned agents. Other approaches propose to minimize the distance between the curriculum distribution and the desired outcome distribution (Ren et al., 2019; Klink et al., 2022), but these require an assumption that the distance between the samples can be measured by the Euclidean distance metric, which cannot be generalized for an arbitrary geometric structure of the environment. Therefore, we argue that the development of algorithms that simultaneously address both outcome-directed exploration and curriculum generation toward the frontier is crucial to benefit from the outcome-directed curriculum RL.

In this work, we propose **O**utcome-directed **U**ncertainty & **T**em**P**oral distance-**A**ware **C**urriculum goal g**E**neration (**OUTPACE**) to address such problems, which requires desired outcome examples only and not prior domain knowledge nor external reward from the environment. Specifically, the key elements of our work consist of two parts. Firstly, our method addresses desired outcome-directed exploration via a Bayesian classifier incorporating an uncertainty quantification based on the conditional normalized maximum likelihood (Zhou & Levine, 2021; Li et al., 2021), which enables our method to propose the curriculum into the unexplored regions and provide directed guidance toward the desired outcomes. Secondly, our method utilizes Wasserstein distance with a time-step metric (Durugkar et al., 2021b) not only for a temporal distance-aware intrinsic reward but also for querying the frontier of the explored region during the curriculum learning. By deploying the above two elements, we propose a simple and intuitive curriculum learning objective formalized with a bipartite matching to generate a set of calibrated curriculum goals that interpolates between the initial state distribution and desired outcome state distribution.

To sum up, our work makes the following key contributions.

- We propose an outcome-directed curriculum RL method which only requires desired outcome examples and does not require an external reward.
- To the best of the author's knowledge, we are the first to propose the uncertainty & temporal distance-aware curriculum goal generation method for geometry-agnostic progress by leveraging the conditional normalized maximum likelihood and Wasserstein distance.
- Through several experiments in goal-reaching environments, we show that our method outperforms the prior curriculum RL methods, most notably when the environment has a geometric structure, and its curriculum proposal shows properly calibrated guidance toward the desired outcome states in a quantitative and qualitative way.

## 2 RELATED WORKS

While a number of works have been proposed to improve the exploration problem in RL, it still remains a challenging open problem. For tackling this problem, prior works include state-visitation counts (Bellemare et al., 2016; Ostrovski et al., 2017), curiosity/similarity-driven exploration (Pathak et al., 2017; Warde-Farley et al., 2018), the prediction model's uncertainty-based

exploration (Burda et al., 2018; Pathak et al., 2019), mutual information-based exploration (Eysenbach et al., 2018; Sharma et al., 2019; Zhao et al., 2021; Laskin et al., 2022), maximizing the entropy of the visited state distribution (Yarats et al., 2021; Liu & Abbeel, 2021a;b). Unfortunately, these techniques are uninformed about the desired outcomes: the trained agent only knows how to visit frontier states as diverse as possible. On the contrary, we consider a problem where the desired outcome can be specified by the given desired outcome examples, allowing for more efficient outcome-directed exploration rather than naive frontier-directed exploration.

Some prior methods that try to accomplish the desired outcome states often utilize the provided success examples (Fu et al., 2018; Singh et al., 2019; Eysenbach et al., 2021; Li et al., 2021). However, they do not provide a mechanism for distilling the knowledge obtained from the agent's experiences into general-purpose policies that can be used to achieve new test goals. In this work, we utilize the Wasserstein distance not only for an arbitrary goal-conditioned agent but also for querying the frontier of the explored region during curriculum learning. Although the Wasserstein distance has been adopted by some previous research, they are often limited to imitation learning or skill discovery (Dadashi et al., 2020; Haldar et al., 2022; Xiao et al., 2019; Durugkar et al., 2021a; Fickinger et al., 2021). Another work (Durugkar et al., 2021b) tries to utilize the Wasserstein distance with the time-step metric for training the goal-reaching agent, but it requires a stationary goal distribution for stable distance estimation. Our work is different from these prior works in that the distribution during training is non-stationary for calibrated guidance to the desired outcome states.

Suggesting a curriculum can also make exploration easier where the agent learns on a tailored sequence of tasks, allowing the agent to autonomously practice the intermediate tasks in a training process. However, prior works often require a significant amount of samples to measure the curriculum's level of difficulty (Florensa et al., 2018; Sukhbaatar et al., 2017), learning progress (Portelas et al., 2020), regret (Jiang et al., 2021). Curricula are often generated by modifying the goal distribution into the frontier of the explored region via maximizing a certain surrogate objective, such as the entropy of the goal distribution (Pong et al., 2019), disagreement between the value function ensembles (Zhang et al., 2020), but these methods do not have convergence mechanism to the desired outcome distribution. While some algorithms formulate the generation of a curriculum as an explicit interpolation between the distribution of target tasks and a surrogate task distribution (Ren et al., 2019; Klink et al., 2022), these still depend on the Euclidean distance metric when measuring the distance between distributions, which cannot be generalized for an arbitrary geometric structure of the environment. In contrast, our method not only provides calibrated guidance to the desired outcome distribution in a sample efficient way but also shows geometry-agnostic curriculum progress by leveraging the bipartite matching problem with uncertainty & temporal distance-based objective.

## 3 PRELIMINARY

We consider the Markov decision process (MDP) $\mathcal{M} = (\mathcal{S}, \mathcal{A}, \mathcal{G}, T, \rho_0, \gamma)$, where $\mathcal{S}$ denotes the state space, $\mathcal{A}$ the action space, $\mathcal{G}$ the goal space, $T(s'|s,a)$ the transition dynamics, $\rho_0$ the initial distribution, $\rho_\pi$ the state visitation distribution when the agent follows the policy $\pi$, and $\gamma$ the discount factor. The MDP in our framework is provided without a reward function and considers an environment in which only the desired outcome sample $\{g_+^k\}_{k=1}^K$ are given, assuming that $g_+^k$ is obtained from the desired outcome distribution $\mathcal{G}^+$. Therefore, our work employs a trainable intrinsic reward function $r : \mathcal{S} \times \mathcal{G} \times \mathcal{A} \to \mathbb{R}$, which is detailed in Section 4.1. Also, we represent the set of curriculum goals as $\{g_c^k\}_{k=1}^N$ and assume that these are sampled from the curriculum goal distribution $\mathcal{G}^c$ obtained by our algorithm.

### 3.1 WASSERSTEIN DISTANCE OVER THE TIME-STEP METRIC

The Wasserstein distance represents how much "work" is required to transport one distribution to another distribution following the optimal transport plan (Villani, 2009; Durugkar et al., 2021b). In this section, we describe the Wasserstein distance over the time-step metric and how it can be obtained with a potential function $f$. Consider a metric space $(\mathcal{X}, d)$, where $\mathcal{X}$ is a set and $d$ is a metric on $\mathcal{X}$, and two probability measures $\mu, \nu$ on $\mathcal{X}$. The Wasserstein-p distance for a given metric $d$ is defined as follows,

$$W_p(\mu,\nu) := \inf_{\gamma \in \Pi(\mu,\nu)} \mathbb{E}_{(X,Y)\sim\gamma}[d(X,Y)^p]^{1/p} \stackrel{\text{if p=1}}{=} \sup_{\|f\|_L \le 1} [\mathbb{E}_{y\sim\nu}[f(y)] - \mathbb{E}_{x\sim\mu}[f(x)]] \quad (1)$$

where a joint distribution $\gamma$ denotes a transport plan, and $\Pi(\mu, \nu)$ denotes the set of all possible joint distributions $\gamma$, and the second equality is held by the Kantorovich-Rubinstein duality with 1-Lipschitz functions ($f : \mathcal{X} \to \mathbb{R}$) (Villani, 2009; Arjovsky et al., 2017).

If we could define the distance metric as $d^\pi(s, s_g)$, which is a time-step metric (quasimetric) based on the number of transition steps experienced before reaching the goal $s_g \in \mathcal{G}$ for the first time when executing the goal-conditioned policy $\pi(a|s, s_g)$, we could design a temporal-distance aware RL by minimizing the Wasserstein distance $W_1(\rho_\pi, \mathcal{G})$ that gives an estimate of the work needed to transport the state visitation distribution $\rho_\pi$ to the goal distribution $\mathcal{G}$:

$$W_1(\rho_\pi, \mathcal{G}) = \sup_{\|f\|_L \leq 1} [\mathbb{E}_{s_g \sim \mathcal{G}}[f(s_g)] - \mathbb{E}_{s \sim \rho_\pi}[f(s)]] \tag{2}$$

Then, the potential function $f$ is approximately increasing along the optimal goal-reaching trajectory. That is, if $\rho_\pi(s)$ consists of the states optimally reaching toward the goal $s_g$, $f(s)$ increases along the trajectory and $f(s_g)$ has the maximum value (Durugkar et al., 2021b;a). Adopting these prior works, the 1-Lipschitz potential function $f$ with respect to $d^\pi(s, s_g)$ could be ensured by enforcing that the difference in values of $f$ on the expected transition from every state is bounded by 1 as follows, (detailed derivations are in Appendix B.)

$$\sup_{s \in \mathcal{S}} \{\mathbb{E}_{a \in \pi(\cdot|s, s_g), s' \in T(\cdot|s, a)}[|f(s') - f(s)|]\} \leq 1. \tag{3}$$

## 3.2 Conditional Normalized Maximum Likelihood (CNML)

For curriculum learning, our work utilizes conditional normalized maximum likelihood (CNML) (Rissanen & Roos; Fogel & Feder, 2018) that can perform an uncertainty-aware classification based on previously observed data by minimizing worst-case regret. Let $\mathcal{D} = \{(s_p, e_p)\}_{p=1}^{n-1}$ be a set of data containing pairs of states $s_{1:n-1}$ and success labels $e_{1:n-1} \in \{0, 1\}$, where '1' represents the occurrence of the desired event. Given a query point $s_n$, CNML in our framework defines the distribution $p_{\text{CNML}}(e_n|s_n)$ which predicts the probability that the state $s_n$ belongs to the desired outcome distribution $\mathcal{G}^+$ ($e = 1$).

To explain how CNML predicts the label of $s_n$, we suppose $\Theta$ is a set of models, where each model $\theta \in \Theta$ can represent a conditional distribution of labels, $p_\theta(e_{1:n}|s_{1:n})$. CNML considers the possibilities that all the possible classes (0,1) will be labeled to the query point $s_n$, and obtains the models $\hat{\theta}(e_{1:n}|s_{1:n}) \in \Theta$ that represent the augmented datasets $\mathcal{D} \cup (s_n, e_n)$ well by solving the maximum likelihood estimation (MLE) problem (LHS of Eq. (4)). Then, CNML that minimizes the regret over those maximum likelihood estimators can be written as follows (Bibas et al., 2019),

$$\hat{\theta}_i = \arg\max_{\theta \in \Theta} \mathbb{E}_{(s,e) \in \mathcal{D} \cup (s_n, e_n = i)}[\log p_\theta(e|s)], \quad p_{\text{CNML}}(e_n = i|s_n) = \frac{p_{\hat{\theta}_i}(e = i|s_n)}{\sum_{j=0}^{1} p_{\hat{\theta}_j}(e = j|s_n)} \tag{4}$$

If the query point $s_n$ is close to one of the data points in the datasets, CNML will have difficulty in assigning a high likelihood to labels that are significantly different from those of nearby data points. However, if $s_n$ is far from the data points in the dataset, each MLE model $\hat{\theta}_{i=0,1}$ will predict the label $e_n$ as its own class, which leads to a large discrepancy in the predictions between the models and provides us with normalized likelihoods closer to uniform (RHS of Eq (4)). Thus, by minimizing the regret through labeling all possible classes to the new query data, CNML can provide a reasonable uncertainty estimate (Li et al., 2021; Zhou & Levine, 2021) on the queried $s_n$ and classify whether the queried $s_n$ is similar to the previously observed data either the forms of label 0, 1, or out-of-distribution data, which is predicted as 0.5.

However, the classification technique via CNML described above is in most cases computationally intractable, as it requires solving separate MLE problems until convergence on every queried data. Previous methods proposed some ideas to amortize the cost of computing CNML distribution (Zhou & Levine, 2021; Li et al., 2021). Following those prior methods, our work adopts MAML (Finn

Figure 2: Visualization of the uncertainty quantification along training progress **(left)** and trained $f_\phi^\pi(s)$ **(right)** in the Point-N-Maze environment. In the **right** figure, high reward means temporally close to the desired outcome states, and low reward means the opposite.

et al., 2017) to address the computational intractability of CNML by training one meta-learner network that can quickly adapt to each model $\hat{\theta}_i$ rather than training each model separately. As Finn et al. (2017) requires samples from $\mathcal{G}^+$ and replay buffer $\mathcal{B}$ for the inference, the probability should be represented as $p_{\text{CNML}}(e = i|s; \mathcal{G}^+, \mathcal{B})$, but we use $p_{\text{CNML}}(e = i|s)$ for notational simplicity in this work. More details about the meta-learning-based classification are included in Appendix B.

## 4 METHOD

For a calibrated guidance of the curriculum goals to the desired outcome distribution, we propose to progress the curriculum towards the uncertain & temporally distant area before converging to $\mathcal{G}^+$, as it is not only the most intuitive way for exploration but also enables the agent to progress without any prior domain knowledge on the environment such as obstacles. In short, our work tries to obtain the distribution of curriculum goals $\mathcal{G}^c$ that are considered (a) temporally distant from $\rho_0$ and, (b) uncertain and, (c) being progressed toward the desired outcome distribution $\mathcal{G}^+$.

### 4.1 TEMPORAL DISTANCE-AWARE RL WITH THE INTRINSIC REWARD

This section details the intrinsic reward for the RL agent as well as the method of training the parameterized potential function $f_\phi^\pi$, trained with the data collected by the policy $\pi(a|s, s_g)$. We consider a 1-Lipschitz potential function $f_\phi^\pi$ whose value increases as the state is far from the initial state distribution and getting close to the goals $s_g \in \mathcal{G}$ proposed by curriculum learning. Then, we can train an agent that reaches the goals $s_g$ in as few steps as possible by minimizing the Wasserstein distance $W_1(\rho_\pi, \mathcal{G})$.

Considering that we can obtain the estimate of $W_1(\rho_\pi, \mathcal{G})$ by Eq (2), the loss for training the parameterized potential function $f_\phi^\pi$ can be represented as follows (Durugkar et al., 2021b;a):

$$\mathcal{L}_\phi = \mathbb{E}_{s,s_g \sim \mathcal{B}}[f_\phi^\pi(s) - f_\phi^\pi(s_g)] + \lambda \cdot \mathbb{E}_{s,s',s_g \sim \mathcal{B}}[\max(|f_\phi^\pi(s) - f_\phi^\pi(s')| - 1, 0))^2] \quad (5)$$

The penalty term with coefficient $\lambda$ in Eq (5) is from Eq (3) for ensuring the smoothness requirement since we consider the Wasserstein distance over the time-step metric. Then, assuming the parameter $\phi$ is trained by Eq (5) at every training iteration, we could obtain the supremum of Eq (2). Thus, the reward can be represented as $r = f_\phi^\pi(s) - f_\phi^\pi(s_g)$, which corresponds to $-W_1(\rho_\pi, \mathcal{G})$.

### 4.2 CURRICULUM LEARNING

As CNML can provide a near-uniform prior (prediction of 0.5) for out-of-distribution data given the datasets (Section 3.2), we could utilize it by treating the desired outcome states in $\mathcal{G}^+$ as $(e = 1)$, and data points in the replay buffer $B$ as $(e = 0)$. Then, we could quantify the uncertainty of a state $s$ based on CNML as

$$\eta_{\text{ucert}}(s, \mathcal{G}^+) = 1 - \left|p_{\text{CNML}}(e = 0|s) - p_{\text{CNML}}(e = 1|s)\right|. \quad (6)$$

which is proportional to the uncertainty of the queried data $s$. However, $\eta_{\text{ucert}}$ alone cannot provide curriculum guidance toward the desired outcome states because it only performs an uninformed search over uncertainties rather than converging to the desired outcome states. Thus, we modify Eq (6) with an additional guidance term:

$$\tilde{\eta}_{\text{ucert}}(\mathcal{G}^c, \mathcal{G}^+) = \mathbb{E}_{s \sim \mathcal{G}^c}[\log(\eta_{\text{ucert}}(s, \mathcal{G}^+) + c \cdot \eta_{\text{guidance}}(s, \mathcal{G}^+))]. \tag{7}$$

where $\eta_{\text{guidance}}(s, \mathcal{G}^+) = (p_{\text{CNML}}(e = 1|s) - 0.5) \cdot \mathbb{1}(p_{\text{CNML}}(e = 1|s) \geq 0.5)$, and $c$ is a hyperparameter that adjusts the preference on the desired outcome states. Since the CNML provides near-uniform prior for out-of-distribution data, $\eta_{\text{ucert}}$ provides large values in the uncertain areas. Also, the guidance term $\eta_{\text{guidance}}(s, \mathcal{G}^+)$ reflects the preference for the states considered to be closer to the desired outcome state distribution.

However, in practice, we found the uncertainty quantification itself sometimes has numerical errors, and it makes $p_{\text{CNML}}$ erroneously predict the states near the initial states or boundaries of the already explored regions as uncertain areas. Therefore, we assume that the curriculum should incorporate the notion of not only the uncertainty but also the temporally distant states from $\rho_0$ for frontier and desired outcome-directed exploration. Thus, we formulate the final curriculum learning objective as follows:

$$\arg\max_{\mathcal{G}^c}[\tilde{\eta}_{\text{ucert}}(\mathcal{G}^c, \mathcal{G}^+)] + L \cdot W_1(\rho_o, \mathcal{G}^c), \tag{8}$$

where the temporal distance bias term with a coefficient $L$ is represented by the Wasserstein distance from initial state distribution $\rho_0$ to the curriculum goal distribution $\mathcal{G}^c$. And, given a parameterized 1-Lipschitz potential function $f_\phi^\pi$ over the time-step metric $d^\pi(s, s_g)$, we can obtain the estimate of $W_1(\rho_o, \mathcal{G}^c)$ by RHS of Eq (1). Also, if we assume $\hat{\mathcal{G}}^c$ to be a finite set of K particles that is sampled from already achieved states in the replay buffer $\mathcal{B}$, the objective function we aim to maximize can be represented as follows:

$$\max_{\hat{\mathcal{G}}^c : |\hat{\mathcal{G}}^c| = K} \sum_{i=1}^{K} [\tilde{\eta}_{\text{ucert}}(s^i, \mathcal{G}^+) + L \cdot [f_\phi^\pi(s^i) - f_\phi^\pi(s_0^i)]], \quad s^i \in \hat{\mathcal{G}}^c, s_0^i \in \rho_0 \tag{9}$$

It enables to propose the curriculum that reflects not only the uncertainty of the states and preference on the desired outcomes but also temporally distant states from $\rho_0$, while not requiring prior domain knowledge about the environment such as an obstacle.

### 4.3 Sampling Curriculum Goal via Bipartite Matching

Since we assume that desired outcome examples from $\mathcal{G}^+$ are given rather than its distribution, we could approximate it by the sampled set $\hat{\mathcal{G}}^+$ ($|\hat{\mathcal{G}}^+| = K$). Then, to solve the curriculum learning problem of Eq (9), we should address the combinatorial setting that requires assigning $\hat{\mathcal{G}}^c$ from the entire curriculum goal candidates in the replay buffer $\mathcal{B}$ to the $\hat{\mathcal{G}}^+$, which is addressed via bipartite matching in this work. With the hyperparameter $c = 4$, we can rearrange Eq (9) as a minimization problem with the costs of cross-entropy loss ($\mathcal{CE}$) and temporal-distance bias ($f_\phi^\pi$) term: (Refer to the Appendix B for the detailed derivation.)

$$\min_{\hat{\mathcal{G}}^c : |\hat{\mathcal{G}}^c| = K} \sum_{s^i \in \hat{\mathcal{G}}^c, g_+^i \in \hat{\mathcal{G}}^+} w(s^i, g_+^i) \tag{10}$$

$$w(s^i, g_+^i) = \mathcal{CE}(p_{\text{CNML}}(e = 1|s^i); y = p_{\text{CNML}}(e = 1|g_+^i)) - L \cdot f_\phi^\pi(s^i) \tag{11}$$

Intuitively, before discovering the desired outcome states in $\mathcal{G}^+$, the curriculum goal $s^i$ is proposed in a region of the state space considered to be uncertain and temporally distant from $\rho_0$ in order to recognize the frontier of the explored regions. And it is kept updated to converge to the desired outcome states for minimizing the discrepancy between the predicted labels of $\hat{\mathcal{G}}^+$ and $\hat{\mathcal{G}}^c$.

Then we can construct a bipartite graph $\mathbf{G}$ with the cost of the edges $w$. Let $\mathbf{V}_a$ and $\mathbf{V}_b$ be the sets of nodes representing achieved states in replay buffer and $\hat{\mathcal{G}}^+$ respectively. We define a bipartite graph $\mathbf{G}(\{\mathbf{V}_a, \mathbf{V}_b\}, \mathbf{E})$ with the weight of the edges $\mathbf{E}(\cdot, \cdot) = -w(\cdot, \cdot)$ and separated partitions ($\mathbf{V}_a$ and $\mathbf{V}_b$). To solve the bipartite matching problem, we utilize the Minimum Cost Maximum Flow

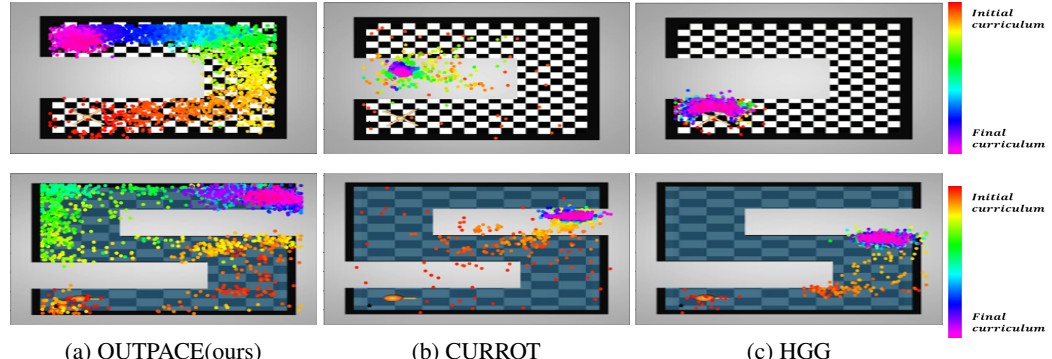

(a) OUTPACE(ours)          (b) CURROT          (c) HGG

Figure 3: Visualization of the proposed curriculum goals. **First row**: Ant Locomotion, **Second row**: Point-N-Maze.

algorithm to find K edges with the minimum cost $w$ connecting $\mathbf{V}_a$ and $\mathbf{V}_b$. (Ahuja et al., 1993; Ren et al., 2019). The overall training process is summarized in Algorithm 1 in Appendix B.

## 5 EXPERIMENTS

We include 6 environments to validate our proposed method. Firstly, various maze environments (Point U-Maze, N-Maze, Spiral-Maze) are used to validate the geometry-agnostic curriculum generation capability. Also, we experimented with the Ant-Locomotion and Sawyer-Peg Push, Pick&Place environments to evaluate our method in more complex dynamics or other domains rather than navigation tasks.

We compare with other previous curriculum or goal generation methods, where each method has the following properties. **HGG** (Ren et al., 2019): Minimize the distance between the curriculum and desired outcome state distributions based on the Euclidean distance metric and value function bias. **CURROT** (Klink et al., 2022): Interpolate between the curriculum and desired outcome state distribution based on the agent's current performance via optimal transport. **GoalGAN** (Florensa et al., 2018): Generate curriculum goals that are intermediate level of difficulty by training GAN (Goodfellow et al., 2014). **PLR** (Jiang et al., 2021): Sample increasingly difficult tasks by prioritizing the levels of tasks with high TD errors. **ALP-GMM** (Portelas et al., 2020): Fit a GMM with an absolute learning progress score approximated by the absolute reward difference. **VDS** (Zhang et al., 2020): Prioritize goals that maximize the epistemic uncertainty of the value function ensembles. **SkewFit** (Pong et al., 2019): Maximize the entropy of the goal distribution to be uniform on the feasible state space via skewing the distribution trained by VAE (Kingma & Welling, 2013).

### 5.1 EXPERIMENTAL RESULTS

Firstly, to show how each module in our method is trained, we visualized the uncertainty quantification by CNML, and $f_\phi^\pi(s)$ values which are proportional to the required timesteps to reach $s$ from $\rho_0$. The uncertainty quantification results (Figure 2) show that the classifier $p_{\mathrm{CNML}}(\cdot|s)$ successfully discriminates the queried states as already explored region or desired outcome states, otherwise, uncertain states. Due to the geometry-agnostic property of the classifier $p_{\mathrm{CNML}}(\cdot|s)$, we could propose the curriculum in the arbitrary geometric structure of the environments, while most of the previous curriculum generation methods do not consider it.

We also visualized the values of the trained potential function $f_\phi^\pi(s)$ to show how the intrinsic reward is shaped (Figure 2). As the potential function $f_\phi^\pi(s)$ is trained to have high values near the desired outcome states due to the Wasserstein distance with the time-step metric, the results show gradual increases of $f_\phi^\pi(s)$ values along the trajectory toward the desired outcome states. That is, high values of $f_\phi^\pi(s)$ indicate the smaller required timesteps to reach the desired outcome state, and this property brings the advantage in identifying the frontier of the explored region.

To validate whether the curriculum goals are properly interpolated from initial states to desired outcome states by combining both objectives for curriculum learning (Eq (8)), we evaluated the

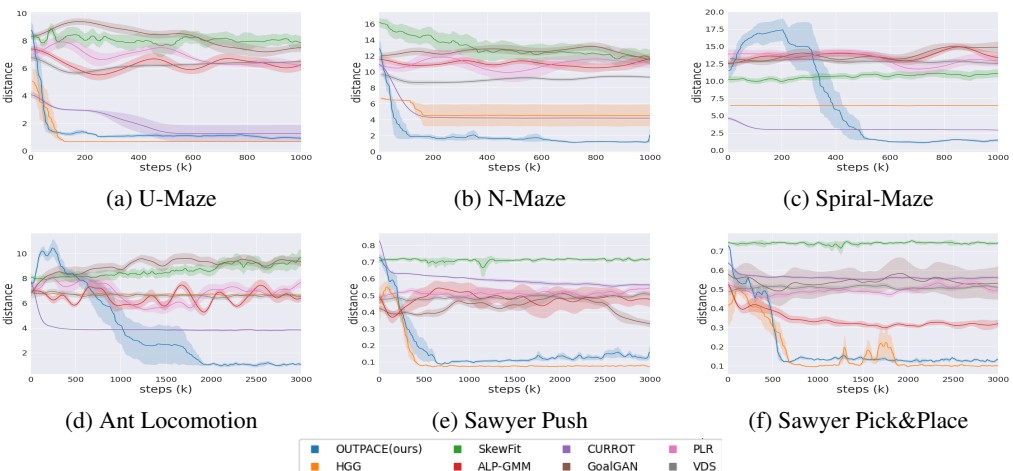

Figure 4: Average distance from the curriculum goals to the final goals (**Lower is better**). Our method's increasing tendencies at initial steps in some environments are due to the geometric structure of the environments themselves. Shading indicates a standard deviation across 5 seeds.

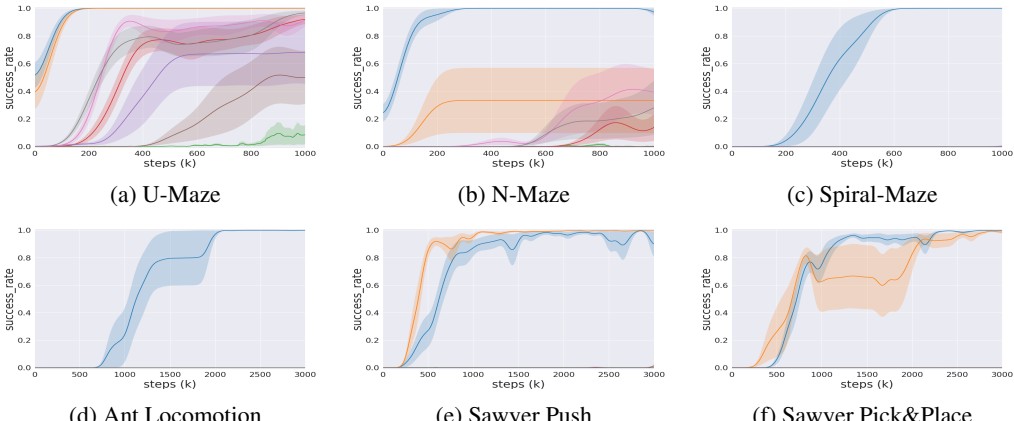

Figure 5: Episode success rates of the evaluation results. The legends and seeds are the same as Fig 4. Note that the curves of the baselines in some environments are not visible as they overlap at zero success rates.

progress of the curriculum goals in a quantitative and qualitative way. For quantitative evaluation, we compare with other previous works described above with respect to the distance from the proposed curriculum goals to $G^+$. As we can see in Figure 4, our method is the only one that consistently interpolates from initial states to $G^+$ as training proceeds, while others have difficulty with complex dynamics or geometry of the environments. For qualitative evaluation, we visualized the curriculum goals proposed by our method and other baselines that show somewhat comparable results as training proceeds (Figure 3). The results show that our method consistently proposes the proper curriculum based on the required timesteps and uncertainties regardless of the geometry and dynamics of the various environments, while other baselines have difficulty as they utilize the Euclidean distance metric to interpolate the curriculum distribution to the $G^+$. We also evaluated the desired outcome-directed RL performance. As we can see in Figure 5, our method is able to very quickly learn how to solve these uninformed exploration problems through calibrated curriculum goal proposition.

## 5.2 ABLATION STUDY

**Types of curriculum learning cost.** We first evaluate the importance of each curriculum learning objective in Eq (8). Specifically, we experimented only with uncertainty-related objective (**only-cnml**) and timestep-related objective (**only-f**) when curriculum learning progresses. As we can see

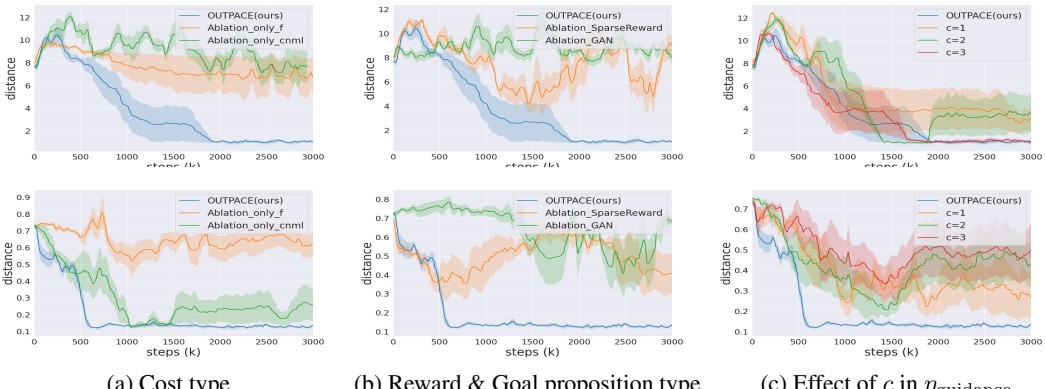

(a) Cost type      (b) Reward & Goal proposition type      (c) Effect of $c$ in $\eta_{\text{guidance}}$

Figure 6: Ablation study in terms of the distance from the proposed curriculum goals to the desired final goal states (**Lower is better**). **First row**: Ant Locomotion. **Second row**: Sawyer Pick & Place. Shading indicates a standard deviation across 5 seeds.

in Figure 6a, both objectives play complementary roles, which support the requirement of both objectives. Without one of them, the agent has difficulty in progressing the curriculum goals toward the desired outcome states due to the local optimum of $f_\phi^\pi$ or the numerical error of $p_{\text{CNML}}$, and more qualitative/quantitative results and analysis about this and other ablation studies are included in Appendix C.

**Reward type & Goal proposition method.** Secondly, we replace the intrinsic reward with the sparse reward, which is typically used in goal-conditioned RL problems, to validate the effects of the timestep-proportionally shaped reward. Also, for comparing the curriculum proposition method, we replace the Bipartite Matching formulation with a GAN-based generative model, which is similar to Florensa et al. (2018), but we label the highly uncertain states as positive labels instead of success rates. As we can see in Figure 6b, the timestep-proportionally shaped reward shows consistently better results due to the more informed reward signal compared to the sparse one, and the generative model has difficulty in sampling the proper curriculum goals because GAN shows training instability with the drastic change of the positive labels, while our method is relatively insensitive because curriculum candidates are obtained from the experienced states from the buffer $\mathcal{B}$ rather than the generative model.

**Effect of $c$ in $\eta_{\text{guidance}}$.** Lastly, we experiment with different values of hyperparameter $c$ to validate the effect of $\eta_{\text{guidance}}$ on curriculum learning. When $c$ is smaller than the default value 4, it is still possible to explore most of the feasible state space except the area near the desired outcome states due to the uncertainty & temporal distance-aware curriculum (Figure 6c). But, we could verify that $\eta_{\text{guidance}}$'s effect becomes smaller as $c$ decreases and $\eta_{\text{guidance}}$ helps to guide the curriculum goals to the desired outcome states precisely. This is consistent with our analysis that the uncertainty & temporal distance themselves can provide curriculum goals in the frontier of the explored region while $\eta_{\text{guidance}}$ can further accelerate the guidance to the desired outcome states.

## 6 CONCLUSIONS

In this work, we consider an outcome-directed curriculum RL where the agent should progress toward the desired outcome automatically without the reward function and prior knowledge about the environment. We propose OUTPACE, which performs uncertainty, temporal distance-aware curriculum RL with intrinsic reward, based on the classifier by CNML, and Wasserstein distance with time-step metric. We show that our method can outperform the previous methods regarding sample efficiency and curriculum progress quantitatively and qualitatively. Even though our method shows promising results, there are some issues with computational complexity due to the innate properties of the meta-learning inference procedure itself. Thus, it would be interesting future work to find a way to reduce the inference time for less training wall-clock time.

## 7 ACKNOWLEDGEMENT

This work was supported by AI based Flight Control Research Laboratory funded by Defense Acquisition Program Administration under Grant UD200045CD. Seungjae Lee would like to acknowledge financial support from Hyundai Motor Chung Mong-Koo Foundation.

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

# A    Training & Experiments details

## A.1    Training details

**Baselines.**    The baseline curriculum RL algorithms are trained as follows,

- HGG (Ren et al., 2019) : We follow the default setting in the original implementation from `https://github.com/Stilwell-Git/Hindsight-Goal-Generation`.
- CURROT (Klink et al., 2022): We follow the default setting in the original implementation from `https://github.com/psclklnk/currot`.
- GoalGAN (Florensa et al., 2018), PLR (Jiang et al., 2021), VDS (Zhang et al., 2020), ALP-GMM (Portelas et al., 2020) : We follow the default setting in implementation from `https://github.com/psclklnk/currot`.
- SkewFit (Pong et al., 2019): We follow the state-based version of SkewFit. The original implementation was modified and used since only the image-based version is provided in it. (`https://github.com/rail-berkeley/rlkit`)

All the baselines are trained by SAC (Haarnoja et al., 2018) with the sparse reward except for the SkewFit as it uses a reward based on the conditional entropy. Even though some algorithms' original implementation is based on on-policy algorithms such as TRPO or PPO (Schulman et al., 2015; 2017), for comparing the sample efficiency, we replace the on-policy algorithm with the off-policy algorithm, SAC, following the referred implementation.

Table 1: Conceptual comparison between our work and the previous curriculum RL algorithms.

| | Uncert. -Aware | Timestep -Aware | Target curriculum dist. | Off- policy | Curriculum proposal | Without ext. reward | Without non- forgetting mechanism |
|---|---|---|---|---|---|---|---|
| HGG | ✗ | ✗ | $\mathcal{G}^+$ | ✓ | $\mathcal{B}$ | ✗ | ✓ |
| GoalGAN | ✗ | ✗ | ✗ | ✗ | GAN | ✗ | ✗ |
| CURROT | ✗ | ✗ | $\mathcal{U}$ or $\mathcal{G}^+$ | ✓ | $\mathcal{U}$ | ✗ | ✗ |
| PLR | ✗ | ✗ | ✗ | ✗ | $\mathcal{B}$ | ✗ | ✗ |
| VDS | ✓ | ✗ | ✗ | ✓ | $\mathcal{B}$ | ✗ | ✓ |
| ALP-GMM | ✗ | ✗ | ✗ | ✓ | GMM | ✗ | ✗ |
| SkewFit | ✓ | ✗ | ✗ | ✓ | VAE | ✓ | ✗ |
| **OUTPACE (ours)** | ✓ | ✓ | $\mathcal{G}^+$ | ✓ | $\mathcal{B}$ | ✓ | ✓ |

We included a conceptual comparison between our work and previous other curriculum or goal generation methods in Table 1.

- **Uncert.-Aware**: whether the curriculum goal proposal process is aware of the uncertainty of the candidate goals.

- **Timestep-Aware**: whether the curriculum goal proposal process is aware of the temporal distance from the initial states or from the desired outcome states.

- **Target curriculum dist**: whether there exists a mechanism for the curriculum goals to converge to the target distribution. When there is no target distribution (e.g. just exploring or expanding the curriculum goal distribution as diverse as possible.), we denoted it as ✗.

- **Off-policy**: whether the off-policy RL can be applied. Some baselines need to measure a kind of difficulty, which means they require repeated trials and on-policy RL algorithms with multi-processing such as TRPO, and PPO (Schulman et al., 2015; 2017).

- **Curriculum proposal**: where the curriculum goals are proposed from.

- **Without ext. reward**: whether the algorithm requires external environmental reward or not.

- **Without non-forgetting mechanism**: whether the algorithm requires implicit or explicit non-forgetting mechanisms. Some baselines mix the previously practiced curriculum goals with a fixed or varying ratio or make the curriculum distribution into uniform over the state space to cover all possible test goal states.

**Training details.**    To train the potential function $f_\phi$ via Eq. (5), $s$ and $s_g$ in the buffer should ideally contain all feasible states in the environment. However, until the policy is learned enough to explore the map, obtaining such ideal distribution is difficult. To mitigate this issue, following Durugkar et al. (2021b), we approximate such a distribution with a small replay buffer $\mathcal{B}_f$ containing recent trajectories and the relabelling technique (Andrychowicz et al., 2017). While this approximation does not provide $f_\phi$ with the ideal state distribution covering all feasible states, we empirically found that this assumption works well since OUTPACE only queries $f_\phi$ for the explored area when it generates curriculum goals.

Table 2: Hyperparameters for OUTPACE

| | | | |
|---|---|---|---|
| Critic hidden dim | 512 | discount factor $\gamma$ | 0.99 |
| Critic hidden depth | 3 | $f_\phi$ update frequency | 1000 |
| Critic target $\tau$ | 0.01 | # of gradient steps for $f_\phi$ update | 10 |
| Critic target update frequency | 2 | # of ensemble networks for $f_\phi$ | 5 |
| Actor hidden dim | 512 | learning rate for $f_\phi$ | 1e-4 |
| Actor hidden depth | 3 | RL optimizer | adam |
| Actor update frequency | 2 | Meta-learner network hidden size | [2048,2048] |
| RL batch size | 512 | Meta-learner train sample size | 512 |
| Init temperature $\alpha_{\text{init}}$ of SAC | 0.3 | Meta-learner test sample size | 2048 |
| Replay buffer $\mathcal{B}$ size | 3e6 | Meta-learner test batch size | 2048 |

Table 3: Env-specific hyperparameters for OUTPACE

| Env name | $\mathcal{B}_f$ size | $\lambda$ for $f_\phi$ | adam $\epsilon$ for SAC | $L$ |
|---|---|---|---|---|
| Point-U-Maze-v0 | 10000 | 25 | 1e-2 | 0.02 |
| Point-N-Maze-v0 | 10000 | 25 | 1e-2 | 0.02 |
| Point-Spiral-Maze-v0 | 20000 | 50 | 1e-8 | 0.02 |
| Ant Locomotion-v0 | 50000 | 25 | 1e-8 | 0.02 |
| Sawyer-Peg-Push | 30000 | 25 | 1e-2 | 2 |
| Sawyer-Peg-Pick&Place | 30000 | 25 | 1e-2 | 0.02 |

## A.2    ENVIRONMENT DETAILS

- Point-U-Maze : The observation consists of the $xy$ position, angle, velocity, and angular velocity of the 'point'. The action space consists of the velocity and angular velocity of the 'point'. The initial state of the agent is $[0, 0]$ and the desired outcome states are obtained by adding uniform noise to the default goal point $[0, 8]$. The size of the map is $12 \times 12$.

- Point-N-Maze : It is the same as the Point-U-Maze environment except that the desired outcome states are obtained by adding uniform noise to the default goal point $[8, 16]$, and the size of the map is $12 \times 20$.

- Point-Spiral-Maze : It is the same as the Point-U-Maze environment except that the desired outcome states are obtained by adding uniform noise to the default goal point $[8, -8]$, and the size of the map is $20 \times 20$.

- Ant Locomotion : The observation consists of the $xyz$ position, $xyz$ velocity, joint angle, and joint angular velocity of the 'ant'. The action space consists of the torque applied on the rotor of the 'ant'. The initial state of the agent is $[0, 0]$ and the desired outcome states are obtained by adding uniform noise to the default goal point $[0, 8]$. The size of the map is $12 \times 12$.

- Sawyer-Peg-Push : The observation consists of the $xyz$ position of the end-effector, the object, and the gripper's state. The action space consists of the $xyz$ position of the end-effector and gripper open/close control. The initial state of the object is $[0.4, 0.8, 0.02]$ and the desired outcome states are obtained by adding uniform noise to the default goal point $[-0.3, 0.4, 0.02]$. We referred to the metaworld (Yu et al., 2020) and EARL (Sharma et al., 2021) environments.

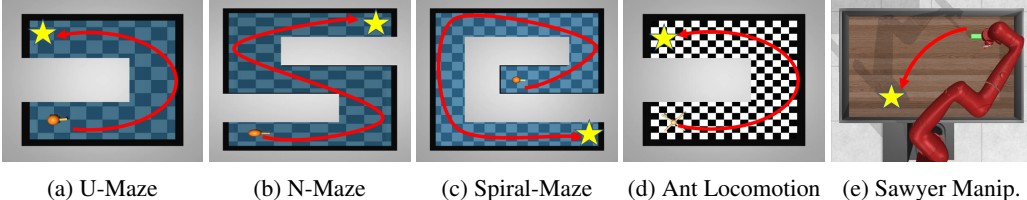

(a) U-Maze  (b) N-Maze  (c) Spiral-Maze  (d) Ant Locomotion  (e) Sawyer Manip.

Figure 7: Environments used for evaluation: **(a)-(c)** the agent must navigate various kinds of maze environments. **(d)** the quadruped ant must navigate the maze to a particular location. **(e)** the robot has to push or pick&place a peg to the desired location.

- Sawyer-Peg-Pick&Place : It is the same as the Sawyer-Peg-Push environment except that the desired outcome states are obtained by adding uniform noise to the default goal point $[-0.3, 0.4, 0.2]$.

## B ALGORITHM & DERIVATIONS

### B.1 ALGORITHM

---
**Algorithm 1** Overview of OUTPACE algorithm

---
1: **Input:** desired outcome examples $\hat{\mathcal{G}}^+$, total training episodes $N$, Env, environment horizon $H$, actor network $\pi$, critic network $Q$, potential function network $f_\phi^\pi$, replay buffer $\mathcal{B}$, $\mathcal{B}_f$
2: **for** iteration=1,2,...,N **do**
3:    $\hat{\mathcal{G}}^c \leftarrow$ sample K curriculum goals that minimize
    $\sum_{s^i \in \mathcal{B}}[\mathcal{CE}(p_{\text{CNML}}(e = 1|s^i); y = p_{\text{CNML}}(e = 1|\hat{\mathcal{G}}^+)) - L \cdot f_\phi^\pi(s^i)]$ (Section 4.3)
4:    **for** $i$=1,2,...,K **do**
5:       Env.reset()
6:       $g \leftarrow \hat{\mathcal{G}}^c$.pop
7:       **for** $t$=0,1,...,$H$-1 **do**
8:          **if** achieved $g$ **then**
9:             $g \leftarrow$ random goal (randomly sample a state with high uncertainty measured by $p_{\text{CNML}}$ in a ball $B_r(s_t)$.)
10:          **end if**
11:          $a_t \leftarrow \pi(\cdot|s_t, g)$
12:          $s_{t+1} \leftarrow$ Env.step($a_t$)
13:       **end for**
14:       $\mathcal{B} \leftarrow \mathcal{B} \cup \{s_0, a_0, s_1...\}$, $\mathcal{B}_f \leftarrow \mathcal{B}_f \cup \{s_0, a_0, s_1...\}$
15:    **end for**
16:    **for** $i$=0,1,...,M **do**
17:       Sample a minibatch b from $\mathcal{B}$ and label reward using $f_\phi^\pi(s_t)$ (Section 4.1)
18:       Train $\pi$ and $Q$ with b via SAC (Haarnoja et al., 2018).
19:       Meta-train $p_{\text{CNML}}$ with b via meta-NML (Algorithm 2.)
20:       Sample a minibatch $b_f$ from $\mathcal{B}_f$
21:       Train $f_\phi^\pi$ with $b_f$ via Eq. (5)
22:    **end for**
23: **end for**

---

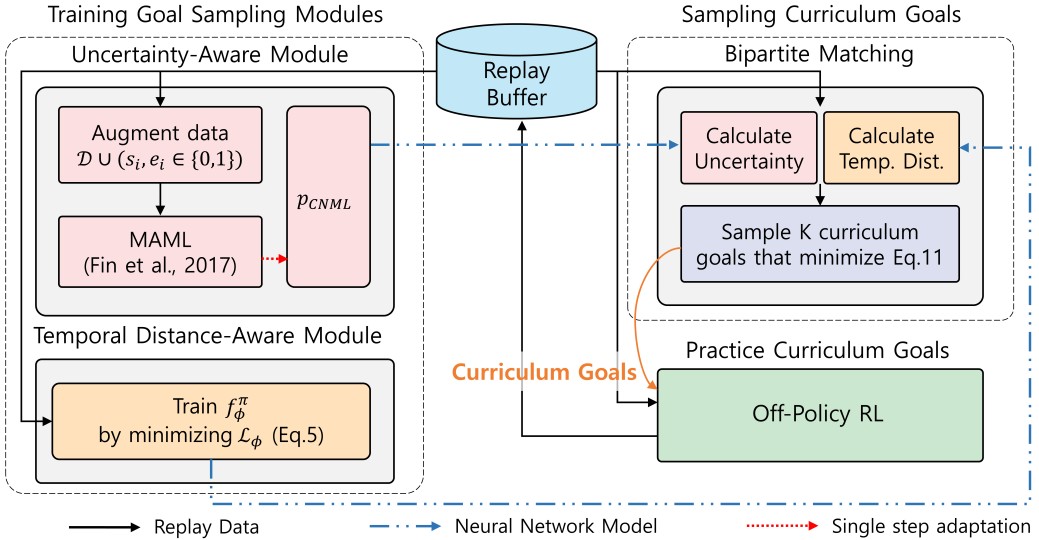

Figure 8: The overall diagram of OUTPACE

---

**Algorithm 2** Meta-NML (Li et al., 2021)

---

1: **Input:** desired outcome examples $\hat{\mathcal{G}}^+$, RL buffer $\mathcal{B}$
2: Sample $\hat{\mathcal{G}}^-$ from RL buffer $\mathcal{B}$
3: $\mathcal{D} \leftarrow \hat{\mathcal{G}}^- \cup \hat{\mathcal{G}}^+$ (Let the size of $\mathcal{D}$ is $n$)
4: Create $2n$ meta-training tasks by relabelling each data points in $\mathcal{D}$ as 0 and 1.
   ($\mathcal{D} \cup (x_i, y')$ where $x_i \sim \mathcal{D}$ and $y' \in \{0, 1\}$)
5: Meta-learn with $2n$ meta-training tasks (Finn et al., 2017)
   (Let $\theta$ be the parameter of $p_{\mathrm{CNML}}$ and $\mathcal{L}$ be standard classification loss.
   $\min_\theta \mathbb{E}_{x_i \sim \mathcal{D}, y' \sim \{0,1\}}[\mathcal{L}(\mathcal{D} \cup (x_i, y'), \theta_{j=y'}^{\prime(i)})]$,
   $s.t. \theta_j^{\prime(i)} = \theta - \alpha \nabla_\theta \mathcal{L}(\mathcal{D} \cup (x_i, y_i), \theta)$

---

### B.2 DERIVATIONS

This section contains the definition of time-step metric $d^\pi$, derivation of Lipschitz smoothness of $f$, and derivation of Eq. (11). Regarding $d^\pi$ and Lipschitz smoothness of $f$, we refer the reader to Durugkar et al. (2021b) for more detailed explanation.

#### B.2.1 TIME-STEP METRIC $d^\pi$, LIPSCHITZ SMOOTHNESS OF $f$

**Definition B.1.** *Given a state space $\mathcal{S}$, action space $\mathcal{A}$, transition dynamics $T : \mathcal{S} \times \mathcal{A} \to \mathcal{S}$, and agent policy $\pi$, the time-step metric $d^\pi$ is a quasi-metric where the distance from $s \in \mathcal{S}$ to $s_g \in \mathcal{S}$ is the expected number of transitions required with the policy $\pi$.*

The time-step metric can be expressed by the expectation of the number of transitions taken under the policy $\pi$, where $\mathcal{T}^\pi(\cdot|s, s_g)$ is the probability distribution of the timestep required to go from $s$ to $s_g$. $d^\pi$ can also be written recursively as

$$
\begin{aligned}
d^\pi(s, s_g) &:= \mathbb{E}_{\pi, \tau \sim \mathcal{T}^\pi(\cdot|s, s_g)}[\tau] \\
&= \begin{cases} 0 & \text{if } s = s_g \\ 1 + \mathbb{E}_{a \sim \pi(\cdot|s, s_g)} \mathbb{E}_{s' \sim T(\cdot|s, a)}[d^\pi(s', s_g)] & \text{otherwise.} \end{cases}
\end{aligned}
\tag{12}
$$

**Lipschitz smoothness of** $f$   If the difference in values of $f$ on the expected transition from every state is bounded by 1, and the policy $\pi$ can reach the goal $s_g$ within a finite number of transitions, then $f$ is the 1-Lipschitz function.

*Proof.* We can write $|f(s_g) - f(s_0)|$ via telescopic sum,

$$
\begin{aligned}
|f(s_g) - f(s_0)| &= \mathbb{E}_{\pi,\tau \sim \mathcal{T}^\pi(\cdot|s_0,s_g)} \left[ \left| \sum_{t=0}^{\tau-1} (f(s_{t+1}) - f(s_t)) \right| \right] \\
&\leq \mathbb{E}_{\pi,\tau \sim \mathcal{T}^\pi(\cdot|s_0,s_g)} \left[ \sum_{t=0}^{\tau-1} |f(s_{t+1}) - f(s_t)| \right].
\end{aligned}
\tag{13}
$$

Since $\mathbb{E}[|f(s') - f(s)|] \leq 1$ by Eq (3), we can write

$$
\begin{aligned}
\mathbb{E}_{\pi,\tau \sim \mathcal{T}^\pi(\cdot|s_0,s_g)} \left[ \sum_{t=0}^{\tau-1} |f(s_{t+1}) - f(s_t)| \right] &\leq \mathbb{E}_{\pi,\tau \sim \mathcal{T}^\pi(\cdot|s_0,s_g)} \left[ \sum_{t=0}^{\tau-1} 1 \right] \\
&= \mathbb{E}_{\pi,\tau \sim \mathcal{T}^\pi(\cdot|s_0,s_g)}[\tau] \\
&= d^\pi(s_0, s_g)
\end{aligned}
\tag{14}
$$

Thus, $f$ is the 1-Lipschitz function with respect to the time-step metric $d^\pi$.  □

### B.2.2   DERIVATION OF EQUATION (11)

By substituting Eq. (7) into Eq (9), and omitting $f_\phi^\pi(s_0^i)$ which is not related to $\hat{\mathcal{G}}^c$, we obtain the following terms,

$$
\begin{aligned}
\max_{\hat{\mathcal{G}}^c:|\hat{\mathcal{G}}^c|=K} \sum_{i=1}^{K} &\Big[ \log \big\{ 1 - \big| p_{\text{CNML}}(e=0|s^i) - p_{\text{CNML}}(e=1|s^i) \big| \\
&+ c(p_{\text{CNML}}(e=1|s^i) - 0.5) \cdot \mathbb{1}(p_{\text{CNML}}(e=1|s^i) > 0.5) \big\} + L \cdot f_\phi^\pi(s^i) \Big], \quad s^i \in \hat{\mathcal{G}}^c.
\end{aligned}
\tag{15}
$$

Also, if we use the default value of the hyperparameters $c = 4$, we can express the above terms as follows,

$$
\begin{aligned}
\max_{\hat{\mathcal{G}}^c:|\hat{\mathcal{G}}^c|=K} \sum_{i=1}^{K} &\Big[ \log \big\{ 1 - \big| 1 - 2p_{\text{CNML}}(e=1|s^i) \big| + \\
&4(p_{\text{CNML}}(e=1|s^i) - 0.5) \cdot \mathbb{1}(p_{\text{CNML}}(e=1|s^i) > 0.5) \big\} + L \cdot f_\phi^\pi(s^i) \Big], \quad s^i \in \hat{\mathcal{G}}^c
\end{aligned}
\tag{16}
$$

which can be simplified as

$$
\min_{\hat{\mathcal{G}}^c:|\hat{\mathcal{G}}^c|=K} \sum_{i=1}^{K} [-\log(p_{\text{CNML}}(e=1|s^i)) - L \cdot f_\phi^\pi(s^i)], \quad s^i \in \hat{\mathcal{G}}^c
\tag{17}
$$

If we could assume that $p_{\text{CNML}}$ is well trained to classify the desired outcome examples $g_+^i \in \hat{\mathcal{G}}_+$ from the states $s$ in the already explored region, $p_{\text{CNML}}(e=1|g_+^i)$ is approximately equal to 1 ($p_{\text{CNML}}(e=1|g_+^i) \approx 1$). Then, the terms inside the above minimization objective can be approximately represented as

$$
-p_{\text{CNML}}(e=1|g_+^i) \log(p_{\text{CNML}}(e=1|s^i)) - L \cdot f_\phi^\pi(s^i).
\tag{18}
$$

Then, in practical implementation, we can implement the above terms by cross-entropy loss as

$$w(s^i, g_+^i) := \mathcal{CE}(p_{\text{CNML}}(e = 1|s^i); y = p_{\text{CNML}}(e = 1|g_+^i)) - L \cdot f_\phi^\pi(s^i), \qquad (19)$$

which is Eq (11). Even though it is not exactly equivalent to the mathematical definition of the cross-entropy, we can just implement it with cross-entropy loss developed in standard deep learning framework such as PyTorch (Paszke et al., 2019) because choosing $s^i$ to maximize $\log(p_{\text{CNML}}(e = 1|s^i))$ and choosing $s^i$ close to $g_+^i$ in order to be classified as desired outcome examples by the $p_{\text{CNML}}$ (predicted labels of $s^i$ to be close to 1) have the same intuitive meaning.

### B.3 A DETAILED DESCRIPTION OF META-NML

Conditional normalized maximum likelihood (CNML) can perform a conservative k-way classification based on previously seen data (Li et al., 2021; Zhou & Levine, 2021). Let $\mathcal{D}_{train} = \{(x_i, y_i)\}_{i=1}^{n-1}$ be a set of data containing pairs of inputs $x_{1:n-1}$ and labels $y_{1:n-1} \in \{1, \cdots, k\}$, where $k$ is the number of possible labels, and $\Theta$ is a set of models. Given a new input $x_n$, CNML defines the distribution $p_{\text{CNML}}(y_n = i|x_n)_{i \in \{1, \cdots, k\}}$ by minimizing the regret $R$ of the worst-case label $y_n$ as

$$p_{\text{CNML}} = \arg\min_q \max_{y_n} R(q, x_{1:n}, y_{1:n}, \Theta), \qquad (20)$$

where we define a regret $R$ for label $y_n$ of a distribution $q$ and maximum likelihood estimator $\theta'$ as

$$R(q, x_{1:n}, y_{1:n}, \Theta) := \log_{\theta'(y_{1:n}|x_{1:n})}(y_{1:n}|x_{1:n}) - \log q(y_{1:n}). \qquad (21)$$

By solving Eq. (20), CNML predicts the distribution of the new label $y_n$ as Eq. (22) (Bibas et al., 2019).

$$p_{\text{CNML}}(y_n = m, x_n) = \frac{p_{\theta_m'^{(n)}}(y = m|x_n)}{\sum_{j=1}^k p_{\theta_j'^{(n)}}(y = j|x_n)}, \qquad (22)$$

where a model $\theta_m'^{(n)}$ is a model that represents the augmented dataset $\mathcal{D} = \mathcal{D}_{train} \cup (x_n, y_n = m)$ well. Thus, the number of total MLE models required is $n \times k$ since we should address each data by augmenting with the label $1, ..., k$, respectively ($\theta_{\beta=1:k}'^{(\alpha=1:n)}$). Since our algorithm utilizes 2-way classification ($k = 2$), we define $2n$ tasks $\tau_{i=1:n}^-$ and $\tau_{i=1:n}^+$, which are constructed by augmenting negative $(x_i, y = 0)$ and positive $(x_i, y = 1)$ labels respectively for each data point $(x_i)$.

To amortize the training cost of the tasks (tasks $\tau_{i=1:n}^{-,+}$) we can apply the meta-learning algorithm (Finn et al., 2017; Li et al., 2021) to this setting and train a model $\theta$ which can quickly adapt to the optimal solution after a single step of gradient update with standard classification loss $\mathcal{L}$ as

$$\min_\theta \mathbb{E}_{x_i \sim \mathcal{D}, y' \sim \{0,1\}}[\mathcal{L}(\mathcal{D} \cup (x_i, y'), \theta_{j=y'}'^{(i)})], \qquad (23)$$

$$s.t. \theta_j'^{(i)} = \theta - \alpha \nabla_\theta \mathcal{L}(\mathcal{D} \cup (x_i, y_i), \theta), \qquad (24)$$

where Eq. (23) and Eq (24) represent the objective of meta-learning and quick adaptation respectively. Training CNML via meta-learning and leveraging CNML are shown in lines 3 and 19 of the algorithm overview (Algorithm 1). Also, we provide the pseudo-code of meta-nml in Algorithm 2.

## C MORE EXPERIMENTAL RESULTS

### C.1 FULL RESULTS OF THE MAIN SCRIPT

We included the full results of the main script in this section. The uncertainty quantification is visualized in Figure 9, the visualization of the trained $f_\phi^\pi(s)$ is in Figure 10, the visualization of

the proposed curriculum goals is in Figure 11. We do not include the visualization results of the Point-U-Maze, and Sawyer-Peg-Pick&Place results as these environments share the same map with the Ant Locomotion, and Sawyer-Peg-Push environments, respectively.

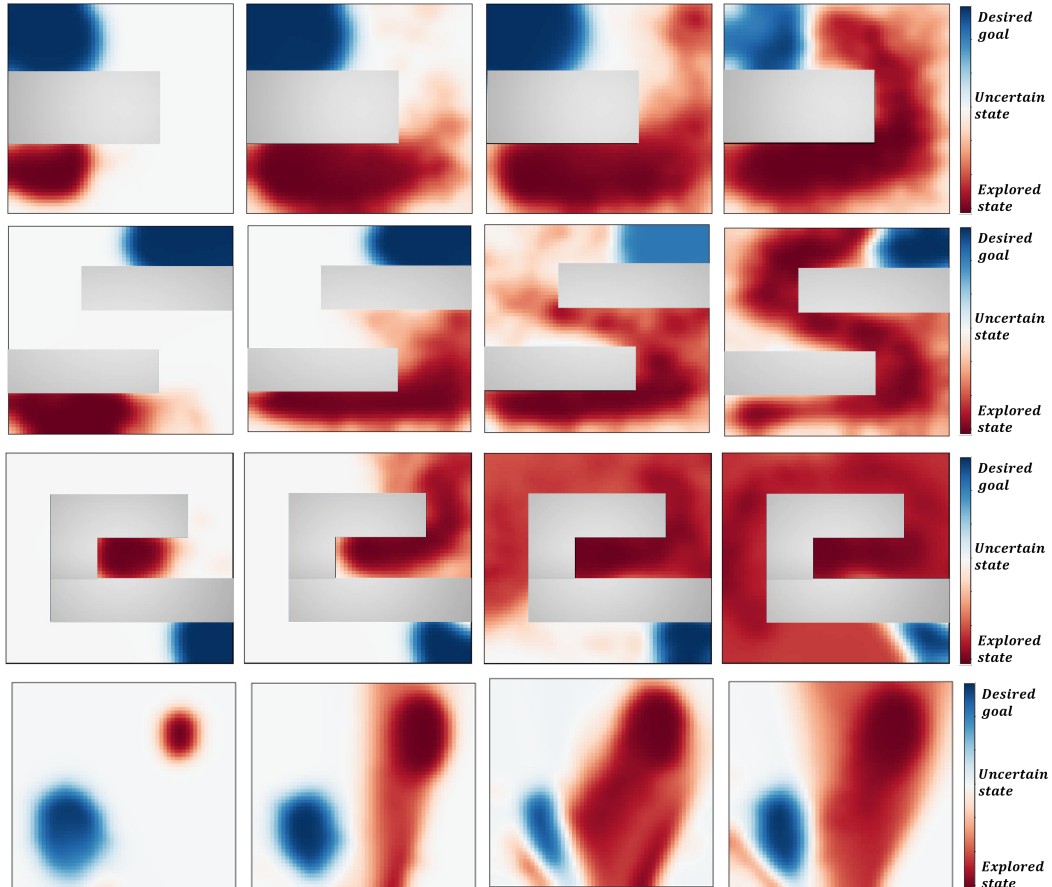

Figure 9: Visualization of the uncertainty quantification along training progress. **First row**: U-Maze (Point, Ant Locomotion), **Second row**: N-Maze, **Third row**: Spiral-Maze, **Fourth row**: Sawyer Peg Push, Pick & Place environments.

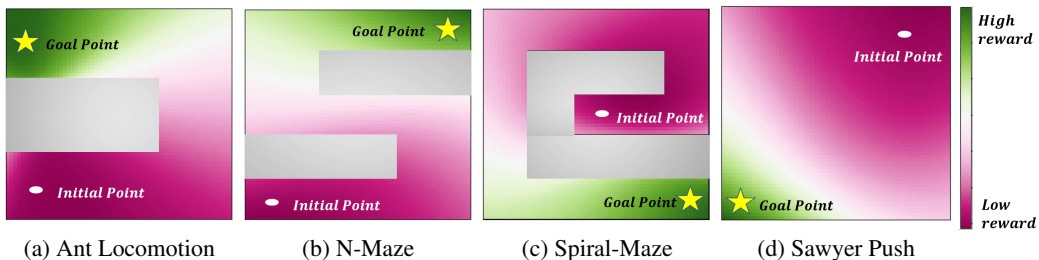

(a) Ant Locomotion     (b) N-Maze     (c) Spiral-Maze     (d) Sawyer Push

Figure 10: Visualization of the trained $f_\phi^\pi(s)$. High reward means temporally close to the desired outcome states (required timesteps are small to reach the goal), and low reward means the opposite (required timesteps are large to reach the goal).

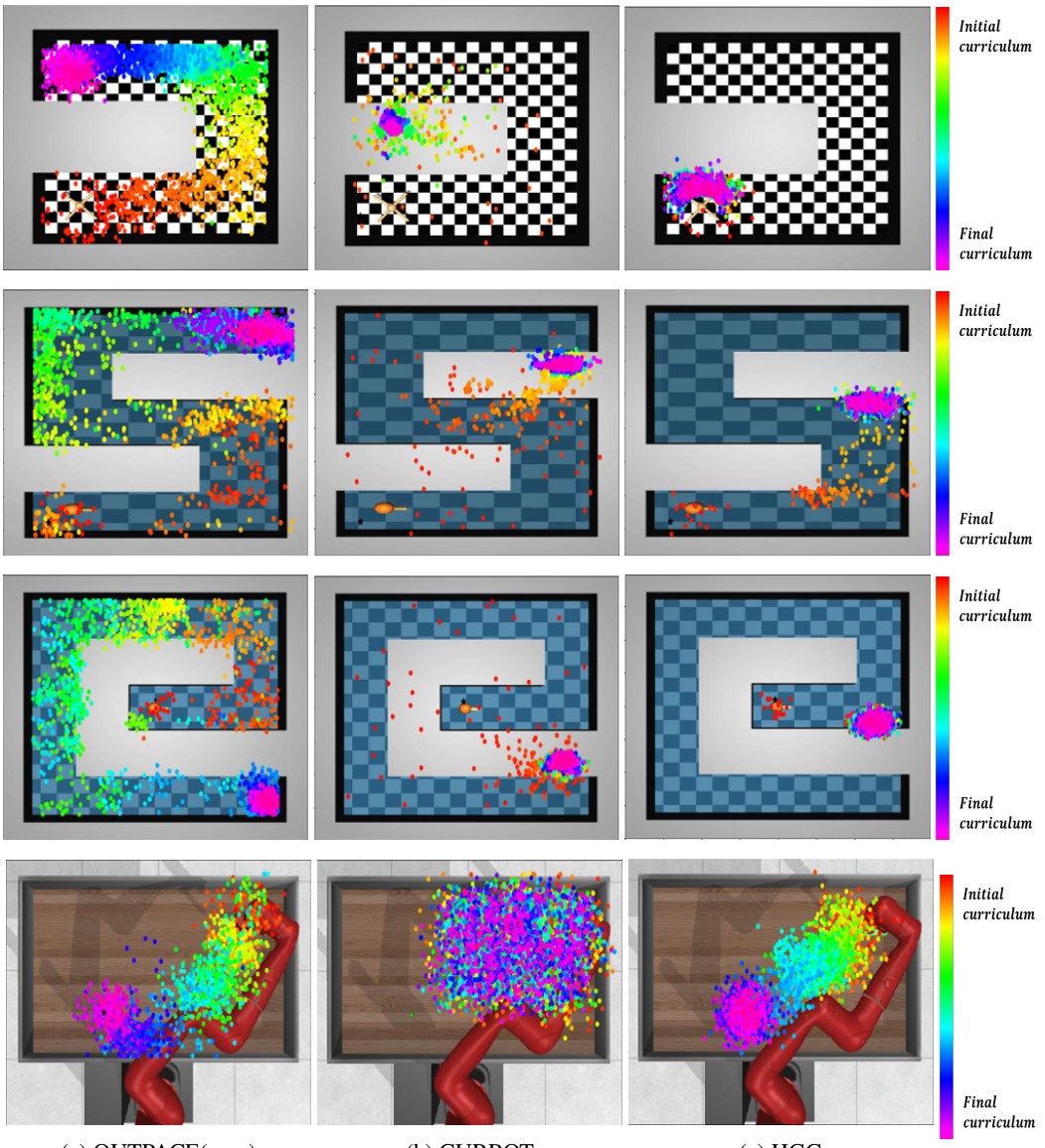

(a) OUTPACE(ours)     (b) CURROT     (c) HGG

Figure 11: Visualization of the proposed curriculum goals. **First row**: Ant Locomotion (same map size with U-Maze), **Second row**: Point-N-Maze, **Third row**: Point-Spiral-Maze, **Fourth row**: Sawyer Peg Push.

## C.2 Evaluation with the goals sampled from the uniform distribution

In some curriculum RL algorithms, they incorporate a mechanism for remembering previously practiced curricula either implicitly or explicitly. For example, **GoalGAN** (Florensa et al., 2018) mixes the previously generated goals with currently generated goals in a specified ratio (e.g. 20 % of previously used goals), and **SkewFit** (Pong et al., 2019) set the objective as targeting the uniform goal distribution (by maximizing the entropy of the goals $\mathcal{H}(g)$), and **CURROT** (Klink et al., 2022) also utilizes uniform target curriculum distribution in practice, and so do some of the other baselines. Due to these algorithmic designs, they require many iterations for explicitly practicing previously used curriculum goals, or show slow progress of curriculum to match the uniform target distribution.

In contrast, our method is based on an intrinsic reward, which is shaped according to the required timesteps proportional values $f_\phi^\pi(s)$ as described in our main script. Thus, our method does not need to explicitly consider the non-forgetting mechanism when we design the algorithm because the reward is already shaped with respect to the timesteps for reaching the arbitrary goal points along the trajectory that reaches the desired outcome state. Because of this property, our method does not consider uniform target distribution or explicitly mixing the previously practiced curriculum goals, and it enables our method to be much faster and show sample-efficient curriculum progress.

We experimented with the goals sampled from the uniform distribution on the feasible state space for validating the previous hypothesis, and the experimental results are shown in Figure 12. Even though our method does not explicitly consider the previously practiced curriculum goals, it shows success in reaching arbitrary goal points sampled from the uniform distribution. Performance degrades are observed in sawyer manipulation environments and these are because we set the uniform distribution as areas within the tables in the environment. But the curriculum goals proposed by our method are converged before the agent explores the entire state space on the table, thus the agent does not have the opportunity to practice the goals from the entire state space.

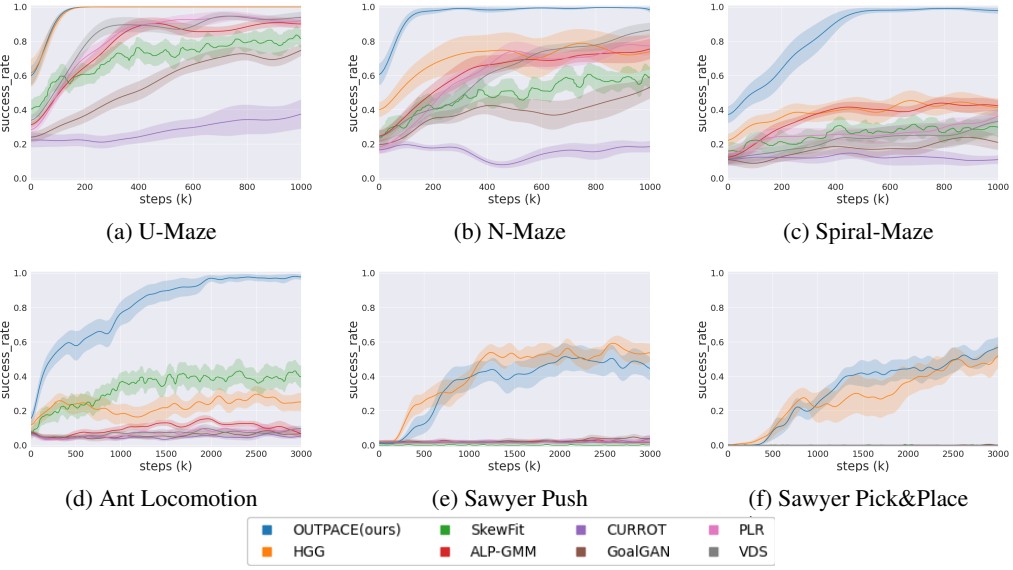

Figure 12: Episode success rates of the evaluation results with the goals uniformly sampled from the feasible state space.

## C.3 Ablation study

We conducted ablation studies described in our main script in all environments. Figure 13 shows the average distance from the proposed curriculum goals to the desired final goal states along the training steps, and Figure 14 shows the episode success rates along the training steps. As we can see in these figures, even though there are not many differences in some environments with simple

dynamics, we could obtain consistent analysis with the results in the main script in most of the environments.

We also visualized the curriculum goals obtained by each ablation study as training proceeds in Figure 15. As we can see in these figures, the curriculum proposal only based on the uncertainty (**only-cnml**) shows progress toward uncertain areas, but it shows unstable curriculum progress, which is depicted by separated curriculum goals despite the same colors or out of order with respect to human intuitive optimal curriculum progress. This is because the meta-learning-based inference procedure of $p_{\text{CNML}}$ has some numerical errors that could lead to the wrong prediction of the states near the boundaries of the already explored regions as uncertain areas (For example, in Figure 9, there exist some states predicted as uncertain area despite already explored region).

Also, even though the curriculum proposal based on the temporal distance only obtained by $f_\phi^\pi$ (**only-f**) shows some progress that deviates from the initial states, it still has difficulty in most of the environments because $f_\phi^\pi$ is trained to reflect the observed transition data rather than entire state space. That is, the temporal distance estimation is reliable within the explored region. Once the trained $f_\phi^\pi$ has local optimum before discovering temporally more distant states (Figure 16), the proposed curriculum goals can be stuck in some areas that are wrongly predicted to be the most far from the initial states in terms of the temporal distance, and it leads to the ineffective exploration of the agent and recurrent failures.

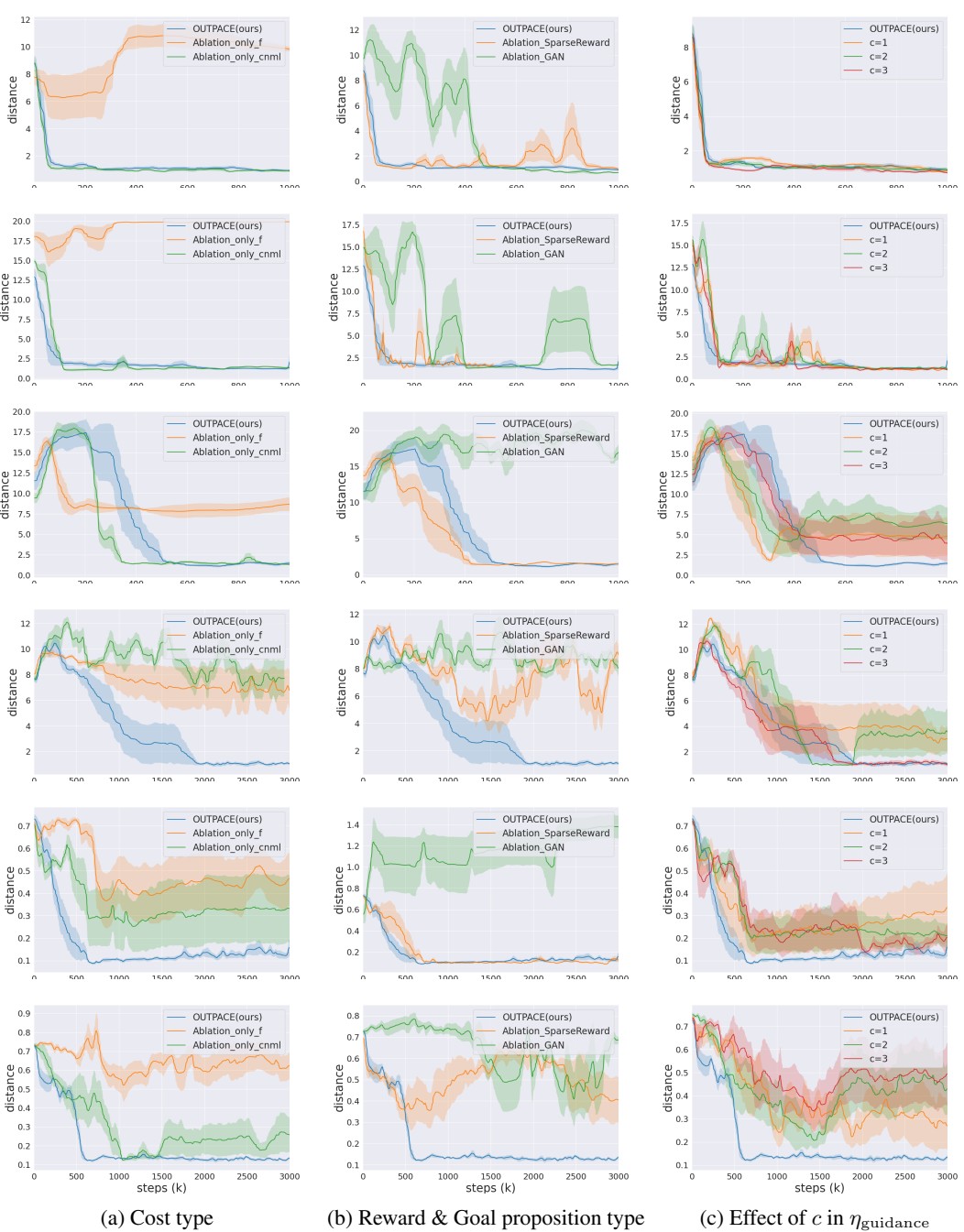

(a) Cost type      (b) Reward & Goal proposition type      (c) Effect of $c$ in $\eta_{\text{guidance}}$

Figure 13: Ablation study in terms of the distance from the proposed curriculum goals to the desired final goal states. **First row**: Point-U-Maze. **Second row**: Point-N-Maze. **Third row**: Point-Spiral-Maze. **Fourth row**: Ant Locomotion. **Fifth row**: Sawyer Push. **Sixth row**: Sawyer Pick&Place. Shading indicates a standard deviation across 5 seeds.

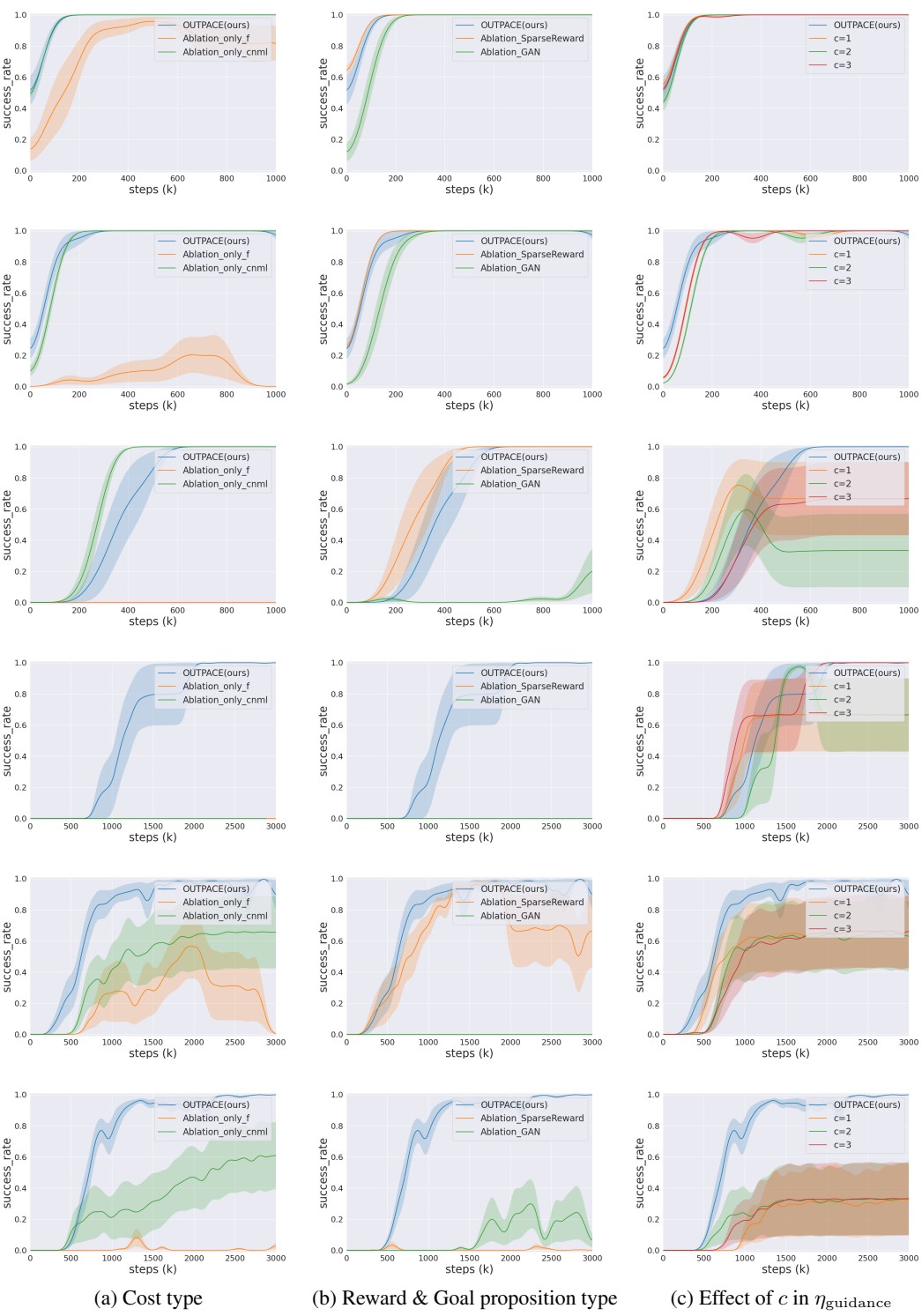

(a) Cost type  (b) Reward & Goal proposition type  (c) Effect of $c$ in $\eta_{\text{guidance}}$

Figure 14: Ablation study in terms of the episode success rates. **First row**: Point-U-Maze. **Second row**: Point-N-Maze. **Third row**: Point-Spiral-Maze. **Fourth row**: Ant Locomotion. **Fifth row**: Sawyer Push. **Sixth row**: Sawyer Pick&Place. Shading indicates a standard deviation across 5 seeds.

(a) OUTPACE(ours)   (b) Ablation: only-cnml   (c) Ablation: only-f

Figure 15: Ablation study in terms of curriculum goals visualization. **First row**: Ant Locomotion, **Second row**: Point-N-Maze, **Third row**: Point-Spiral-Maze, **Fourth row**: Sawyer Manipulation.

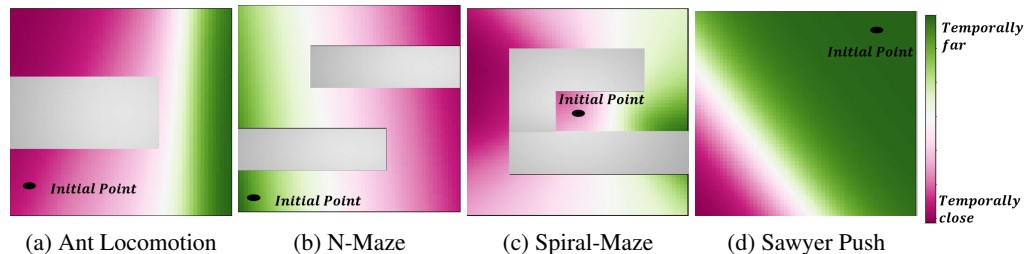

(a) Ant Locomotion   (b) N-Maze   (c) Spiral-Maze   (d) Sawyer Push

Figure 16: Visualization of the **not properly trained** $f_\phi^\pi(s)$ when ablation study **only-f** is performed.

