# OpenReview forum: "Outcome-directed Reinforcement Learning by Uncertainty \& Temporal Distance-Aware Curriculum Goal Generation"
_ICLR.cc/2023/Conference — ICLR 2023 notable top 25%_

### Official Review · Reviewer_J8za · 2022-10-23

**Confidence:** 4
**Correctness:** 3
**Technical Novelty And Significance:** 3
**Empirical Novelty And Significance:** 3
**Recommendation:** 8

**Clarity, Quality, Novelty And Reproducibility:**

- The presented approach is novel, with results demonstrating improved performance over prior methods.
- The experiments are thorough, and the ablation study also validates their design choices.

**Strength And Weaknesses:**

Strengths:
+ Novel approach for curriculum goal generation
+ Detailed experiments with strong baseline methods
+ Ablation study demonstrating the role of both uncertainty and temporal distance on performance.

Weaknesses:
- No evaluation or discussion on the generalization capacity of the proposed method across different environments with unseen start and goals.
- Does the proposed method scale to Sawyer Pick/Place task in environments with obstacles?



**Summary Of The Paper:**

The paper proposes an uncertainty & temporal distance-aware curriculum goal-generation method for various navigation tasks and a manipulation task. Uncertainty quantification is based on a Bayesian classifier obtained via conditional normalized maximum likelihood (Zhou & Levine, 2021; Li et al., 2021), which guides the curriculum into unexplored regions. The temporal distance is represented using a Wasserstein distance with a time-step metric (Durugkar et al., 2021b) to provide intrinsic reward and query the frontier of the explored region. The proposed method is compared against a variety of existing approaches.

**Summary Of The Review:**

Overall, the presented approach is novel and presents thorough comparative and ablative studies. However, it is suggested that authors comment on the generalization capacity of their proposed method across novel settings and of manipulation under collision and other task constraints.

---

> ### Author Response · Authors · 2022-11-11
> **Response to reviewer J8za**
>
> Dear reviewer J8za,
>
> We sincerely appreciate your positive and insightful comments. We found them extremely helpful in improving our manuscript. We prepared our response below:
>
> - - -
>
> **Q1. No evaluation or discussion on the generalization capacity of the proposed method across different environments with unseen start and goals.**
>
> **A1.** Firstly, we sincerely thank you for reviewing our paper and providing constructive comments with pleasure.
>
> Start point : Our work is based on a general goal-conditioned RL framework, which is built on top of the Markov assumption. Thus, the trained policy $\pi(a|s,g)$ is only dependent on the current state. Also, as OUTPACE is designed to explore the environment based on uncertainty quantification, the agent is encouraged to cover the state space as much as possible before discovering the desired outcome states. As all the experiences are obtained in a process of reaching the desired outcome states, with a proper exploration process to cover the state space, the trained policy can reach the desired outcome states from arbitrary initial states in the environment.
>
> Goal point : We think we addressed your comments about the arbitrary goal points in Appendix C.2. We politely ask you to refer to the appendix that shows the evaluation results with uniformly sampled goals from feasible state space (Figure 12: [Click here](https://drive.google.com/file/d/12t3fzWu4rIO-4bZdIgpVyl62XBd_kF8F/view?usp=sharing)). Specifically, after the training process, we evaluated uniformly sampled goals from feasible state space. It shows that OUTPACE can reach the arbitrary given goals, indicating that the agent does not forget about the previously practiced curriculum goals.
>
>
> In summary, the questions about the random start points are addressed in our experiment in Figure 4-5 by Markov assumption. Also, the experiments on random goal points are reported in Appendix C.2. If this response is not sufficient enough to address your comments, please do not hesitate to let us know.
>
> - - -
>
>
> **Q2. Does the proposed method scale to Sawyer Pick/Place task in environments with obstacles?**
>
> **A2.** Thank you for your interesting idea. To address your comment, we additionally experiment with robot manipulation tasks (Sawyer Push, Pick\&Place) with an obstacle. In both environments, we add the rectangular obstacle at the center of the table to induce the agent to detour the obstacle for reaching the desired outcome state.
>
> Sawyer Push with the obstacle :
> https://drive.google.com/file/d/1AaOcN2zPvRTgFCTyyaaQ7i1lP6CEISe4/view?usp=sharing
>
> Sawyer Pick\&Place with the obstacle :
> https://drive.google.com/file/d/1rB4omDB6r-jo9QBe8OVSvxgUpZAoV-33/view?usp=sharing
>
> In the Sawyer Push environment, the agent should detour the obstacle by pushing the object to reach the desired outcome state. In the Sawyer Pick\&Place environment, the agent should detour the obstacle after picking the object as there is no route for reaching the desired outcome state without picking the object.
>
> The results are shown in the following figures:
>
> **Sawyer Push with the obstacle**
>
> Average distance from the curriculum goals to the final goals :
> https://drive.google.com/file/d/1fiWlhySEebiuI_v0TOTfy3Pb9k9bu4IT/view?usp=sharing
>
> episode success rates :
> https://drive.google.com/file/d/1yKVH6izni1lGOXOIheEzfneCgLekmJdD/view?usp=sharing
>
> **Sawyer Pick \& Place with the obstacle**
>
> Average distance from the curriculum goals to the final goals :
> https://drive.google.com/file/d/1l3RKQ1oCFDptj5PYYXmikQqpmjRwvb7c/view?usp=sharing
>
> episode success rates :
> https://drive.google.com/file/d/1bqIZ78dG3ItcvS31C-Pq-z1Vhoqv5yEd/view?usp=sharing
>
> We found that OUTPACE can also be generalized to robotic manipulation tasks with an obstacle, which is consistent with our analysis in the main script. If this response is not sufficient enough to address your comments, please do not hesitate to let us know.
>
>
> - - -
>
> Thank you again for your valuable and insightful review.
>
> Please let us know if our responses have addressed your comments. If anything needs further clarification, please do not hesitate to let us know.
> - - -

---

> > ### Comment · Reviewer_J8za · 2022-11-17
> > **Thank you for your response**
> >
> > Thank you for addressing my comments. Can you share videos of Sawyer Push with the obstacle and Sawyer Pick & Place with the obstacle?

---

> > > ### Author Response · Authors · 2022-11-18
> > > **Thank you for your response**
> > >
> > > We are happy to hear that our response addressed your comments.
> > >
> > > Following your suggestion, we would like to share videos of Sawyer Push with the obstacle and Sawyer Pick & Place with the obstacle.
> > >
> > > Sawyer Push:
> > > https://drive.google.com/file/d/1gAEo-ZmiSOuubhFpGOksiuyJW4e5NGk7/view?usp=sharing
> > >
> > > Sawyer Pick & Place:
> > > https://drive.google.com/file/d/1KCH-8c8Z6AMfRATHN6UwmV3j5r4WOoqi/view?usp=sharing
> > >
> > > We appreciate again your valuable suggestions and all your efforts during the review process.

---

> > > > ### Comment · Reviewer_J8za · 2022-11-18
> > > > **Thank you for addressing my comments**
> > > >
> > > > Thank you. I have no further questions.

---

### Official Review · Reviewer_KAqi · 2022-10-25

**Confidence:** 4
**Correctness:** 4
**Technical Novelty And Significance:** 3
**Empirical Novelty And Significance:** 3
**Recommendation:** 8

**Clarity, Quality, Novelty And Reproducibility:**

The method is clearly written and easy to follow. The work provides a novel algorithm for

**Details Of Ethics Concerns:**

N/A.

**Strength And Weaknesses:**

Strengths
* The method is well-motivated to obtain three desired properties: temporal- and uncertainty-awareness, and closeness to the desired outcome distribution. These properties are empirically verified by experiments.
* The method provides a novel bipartite-matching-based algorithm as an instantiation of the goal sampling process.
* It is empirically demonstrated that the proposed method achieves better asymptotic performance and sample efficiency compared to prior methods.

Weaknesses
* On Spiral-Maze and Ant Locomotion environments, none of the baseline methods are able to have non-zero success rate. The paper does not offer an explanation what part of the proposed algorithm results in such performance discrepancy. Is the discrepancy caused by particular attributes of the environments, and are these attributes taken care of by specific components of the proposed method? A further ablation study of the method design on these environments would provide more intuition.

Clarification questions
* In Figure 3, HGG has its final curriculum centered in a region which is far from the designed goal for U-Maze, while still having a good performance. Why is it the case?

**Summary Of The Paper:**

The paper proposes a curriculum learning framework which generates training goals and demonstrates empirical sample efficiency compared to existing method, on a range of navigation and manipulation tasks.

**Summary Of The Review:**

The method is overall well-motivated by desired properties of intrinsic reward and desired distribution of training goals, with strong empirical performance compared to prior works. It also offers descent amount of ablations and visualizations for further explanations.

---

> ### Author Response · Authors · 2022-11-11
> **Response to reviewer KAqi**
>
> Dear reviewer KAqi,
>
> We sincerely appreciate your positive and insightful comments. We found them extremely helpful in improving our manuscript. We prepared our response below:
> - - -
> **Q1. On Spiral-Maze and Ant Locomotion environments, none of the baseline methods are able to have non-zero success rate. The paper does not offer an explanation what part of the proposed algorithm results in such performance discrepancy. Is the discrepancy caused by particular attributes of the environments, and are these attributes taken care of by specific components of the proposed method? A further ablation study of the method design on these environments would provide more intuition.**
>
> **A1.** Firstly, we sincerely thank you for reviewing our paper and providing constructive comments with pleasure.
>
> First of all, we note that the success rates in Figure 5 show the evaluation results when the agent needs to reach the desired outcome states (final goal) rather than arbitrary goals in the feasible state space. Therefore, for a non-zero success rate, the exploration of the overall state space including the desired outcome states is required first. That is, zero success rates in some baselines are attributed to not enough exploration and curriculum goal progress toward the desired outcome states. If your comments are about the zero success rate itself, we politely ask you to refer to the appendix that shows the evaluation results with uniformly sampled goals from feasible state space (Figure 12 : [Click here](https://drive.google.com/file/d/12t3fzWu4rIO-4bZdIgpVyl62XBd_kF8F/view?usp=sharing)).
>
> If your comments are about the detailed explanation or analysis about the discrepancies, we could think of the reasons in terms of two aspects.
>
> Firstly, in Ant Locomotion \& Point-U-Maze environments, even though they share the same size and shape of the maze, performance discrepancies occur due to the complexity of the agent's dynamics. As we mentioned in the introduction part of the main script, proposing the curriculum goal to the agent is intimately connected to the efficient desired outcome-directed exploration and vice versa. Thus, without the complementary role of both curriculum proposal \& efficient exploration, the agent cannot make progress and fails to reach the desired outcome states.
>
> Due to the relatively simple dynamics compared to the Ant in the Point-U-Maze environment, there is a high possibility that the agent will visit states far from the explored region or close to the desired outcomes by chance, even though the proposed curriculum goal is not one of these states. These unexpectedly visited states can be better curriculum goal candidates, which leads to the progress of the curriculum goals toward the desired outcome states.
>
> However, in the Ant Locomotion environment, there is a very low possibility that the agent visits these unseen states without properly calibrated curriculum goals, because the alternating process of proposing curriculum goals and exploring the environment based on these curriculum goals fails once one of these processes does not work well.
>
> Secondly, in Point-Maze (U, N, Spiral), the difference between these environments is only the geometric shape of the maze. Thus, the discrepancies are attributed to the uncertainty & temporal distance-aware properties of OUTPACE. As other baselines are not geometry-agnostic due to the Euclidean distance metric or the properties that are susceptible to focusing on infeasible goals (e.g. : inside the obstacle), the agent’s capability stagnates in the intermediate level of difficulty. On the contrary, OUTPACE works well due to both geometry-agnostic properties and independency from the level of difficulty, which prevent the agent from repeatedly trying to reach the proposed infeasible curriculum goals.
>
> If this response is not sufficient enough to address your comments, please do not hesitate to let us know.
>
> - - -
>
> **Q2. In Figure 3, HGG has its final curriculum centered in a region which is far from the designed goal for U-Maze, while still having a good performance. Why is it the case?**
>
> **A2.** Firstly, we would like to make clear that HGG’s curriculum goals in the first row of Figure 3 are results of the *Ant-Locomotion* environment, not Point-U-Maze. HGG’s curriculum goal visualization in the Point-U-Maze environment is shown in the following figure
>
> https://drive.google.com/file/d/1nojdkgq6wKI6WUQehCv6sLt_CX9XNkb4/view?usp=sharing
>
> and it shows that converged curriculum goals are at the desired outcome states, which corresponds to the good performance of HGG in Point-U-Maze. If this response is not sufficient enough to address your comments, please do not hesitate to let us know.
>
>
> - - -
>
> Thank you again for your valuable and insightful review.
>
> Please let us know if our responses have addressed your comments. If anything needs further clarification, please do not hesitate to let us know.
> - - -

---

> > ### Comment · Reviewer_KAqi · 2022-11-30
> > **Thank you for your responses**
> >
> > Dear authors,
> >
> > Thank you for your responses and I have no further questions.

---

### Official Review · Reviewer_4feY · 2022-10-26

**Confidence:** 2
**Correctness:** 3
**Technical Novelty And Significance:** 3
**Empirical Novelty And Significance:** 3
**Recommendation:** 6

**Clarity, Quality, Novelty And Reproducibility:**

The writing of this paper can be improved with formal language and grammatically correct and shorter sentences. Some of the descriptions can be confusing and hard to access.
The presentation of both the method and results can also be improved. A visual of the algorithm flow would be super useful.
------------------------

EDIT:  The authors made a good effort addressing my confusions and I do like the technical contribution of this work so I am raising my score but would truly appreciate the authors to further improve the writing of this work.

**Strength And Weaknesses:**

Strength:
-Goal-based curriculum RL is a powerful paradigm of RL that finds policy based on a few examples of goal states by generating gradually challenging goals.
-This work proposes a curriculum generation method that combines uncertainty measure and a temporal-distance function to explore the frontiers of currently explored states.
The intrinsic reward formulation is intuitively and conceptually correct.
-Experiments in simulated domains demonstrate OUTPACE does outperform other curriculum-RL approaches and the ablation study is extensive and thorough.

Weakness:
-The presentation of this work in general needs improvement. The language used is not formal and at the same time not easy to understand with grammatical errors.
-A lot of the technical details are missing from the main text and deferred to the appendix or reference work, which makes understanding the technical contents really difficult. For example, how did you make the CNML estimation tractable? How long or how much longer does the proposed method require compared with baseline methods? How much did the meta learning part help?
-I also have a hard time understanding what the hyperparameter c represents (how does it adjust the preference?) and how bipartite matching actually applies here. A diagram/visual of different moving parts of the algorithm would be super helpful.
-More explanation and analysis of the results could be very useful for understanding the performance of the proposed method. In figure5, it seems a lot of baseline data are missing in many task domains but it is unclear whether there is no data or the success rates are zeros for all of them?


**Summary Of The Paper:**

This work presents a method for proposing curriculum goals given a set of desired goal states for curriculum reinforcement learning. The core idea is to find states at the frontier that are similar to the specified goals. The proposed method keeps track of exploration frontiers using a notion of temporal distance and computes state similarity using an estimated wasserstein distance. Experiments in simulated control tasks show the proposed method outperform baselines.



**Summary Of The Review:**

Interesting work on curriculum RL with seemingly strong result, but the presentation of the work can be hard to access.

---

> ### Author Response · Authors · 2022-11-11
> **Response to reviewer 4feY (1/2)**
>
>
> Dear reviewer 4feY,
>
>
> We sincerely appreciate your constructive and insightful comments. We found them extremely helpful in improving our manuscript. We prepared our response below:
> - - -
> **Q1. Technical details are missing from the main text and deferred to the appendix or reference work. (For example, how did you make the CNML estimation tractable?)**
>
> **A1-1.** Thank you for your suggestion to move the technical details from the Appendix to the main text. Following your suggestion, we moved as many technical details to the main text as the page limit allows (e.g. the name of the meta-learning method (MAML [1]) that makes the CNML estimation tractable). Please pardon that we could not include all of the explanations in the main text due to the page limit.
>
>
> **A1-2.** Also, we would like to explain the details of the ‘meta-learning approach to address computational intractability of CNML’ here, which are included in Appendix B.1 and B.3.
>
>
> It is computationally intractable to classify states as “uncertain” and “certain” via CNML since it requires individual MLE models twice as much as the number of data points in the dataset. To make the CNML estimation tractable, we adopt MAML (Model-Agnostic Meta-Learning) which can amortize the cost of calculating CNML distribution.
>
>
>
> Let $\theta_j'^{(k)}\in\Theta$ ( $j\in\{0,1\}$ ) be the set of MLE models for datasets $\mathcal{D} = (x_k,y_k)\_{k=1}^{n-1} \cup (x_n,y_n)$. Thus, the number of total MLE models required is $2n$ since we should address each data by augmenting with the negative label 0, and positive label 1, respectively. In the meta-learning setting for CNML, we train a meta-learner network $\theta$ that can quickly adapt to each MLE model, rather than train $2n$ different neural networks $(\theta\_{j=0}'^{(k)}, \theta\_{j=1}'^{(k)}\)\_{k=1}^{n}$ separately. To do so, we follow the gradient-based meta-learning objective in MAML (Eq. 23,24, $\mathcal{L}$ is the standard classification loss) to train the meta-learner network $\theta$. When we try to compute CNML, we conduct a single step of gradient update to obtain each MLE model $(\theta\_{j=0}'^{(k)}, \theta\_{j=1}'^{(k)}\)\_{k=1}^{n}$ and use the MLE models to obtain $p_{\mathrm{CNML}}(y=i|x_k) = \frac{p_{\theta_i'^{(k)}}(y=i|x_k)}{\sum\limits\_{j=0}^{1}p\_{\theta_j'^{(k)}}(y=j|x_k)}$.
>
> $$\min_\theta \mathbb{E}_{x_k\sim\mathcal{D},y'\sim\{0,1\}}[\mathcal{L}(\mathcal{D}\cup(x_k,y'),\theta\_{j=y'}'^{(k)})]\quad (Eq.23)$$
>
> $${s.t. \theta_j'^{(k)} = \theta - \alpha \nabla_\theta\mathcal{L}(\mathcal{D}\cup(x_k,y_k),\theta)}\quad (Eq.24)$$
>
> Please let us know if there is anything else that we can revise to promote the reader’s understanding.
>
> [1] Finn, Chelsea, Pieter Abbeel, and Sergey Levine. "Model-agnostic meta-learning for fast adaptation of deep networks." International conference on machine learning. PMLR, 2017.
> - - -
>
>
> **Q2. How long or how much longer does the proposed method require compared with baseline methods?**
>
> **A2.** OUTPACE(ours) takes longer time in the overall training process than other baselines  [e.g. 7 hours (Ant Locomotion) / 11.5 h (Sawyer Push) / 7 h (PointSpiralMaze) longer than baselines for 1M timesteps using an NVIDIA RTX A5000 and AMD Ryzen 2950x]. However, we would like to inform you that OUTPACE consumes fewer environmental steps while other baselines require much more steps or could not explore complex environments at all. Also, the computational complexity of our method is only required during training and it is not required during the evaluation process.
>
>
> - - -
>
> **Q3. How much did the meta learning part help?**
>
> **A3.** Since meta-learning allows quick adaptations to obtain multiple MLE models by maintaining only one meta-learner model, the number of model parameters is approximately reduced by 1/2n where n is the number of data points (n=the capacity of the replay buffer in our framework). In addition, since training each of the 2n neural network models can be simplified by training one meta-learner with adaptation overhead, the training time for the uncertainty module can be reduced (O(n) → O(1)).
>
> - - -

---

> > ### Author Response · Authors · 2022-11-11
> > **Response to reviewer 4feY(2/2)**
> >
> > **Q4. What the hyperparameter c represents? (how does it adjust the preference on the desired outcome states?)**
> >
> > **A4.** $c$ in Eq. 7 is a hyperparameter that determines how much $\tilde{\eta}\_{\mathrm{ucert}}$ reflects $\eta_{\mathrm{guidance}}$,
> >
> > $$\tilde{\eta}\_\mathrm{ucert} = \mathbb{E}[\log(\eta_\mathrm{ucert} + c \cdot \eta_\mathrm{guidance})]\quad (Eq.7.),$$
> >
> > where a large value of $\eta_{\mathrm{guidance}}$ indicates that the value of $p_{\mathrm{CNML}}(e=1|s)$ is close to 1 ( $i.e.$ the CNML classifies the state $s$ as a desired goal), and a large value of $\eta_{\mathrm{ucert}}$ indicates that $s$ is an uncertain state ( $i.e.$ out-of-distribution data). Since the objective function (Eq. 9) we aim to maximize includes $\tilde{\eta}\_\mathrm{ucert}$, $c$ is a hyperparameter that determines how much $\eta_{\mathrm{guidance}}$ affects the goal sampling method. If $c$ has a small value, the goal sampling method prefers uncertain goals to desired goals, and in the opposite case, it prefers desired goals to uncertain goals.
> >
> > The difference according to $c$ is evident when the achieved-goal distribution's support covers the desired-goal distribution. If $c$ has a small value, the curriculum goal proposal method still pursues the uncertain states even though the agent already explored the desired goal area. When $c$ is large enough, the agent is instructed to practice the desired goal intensively.
> >
> > - - -
> >
> > **Q5. How bipartite matching actually applies here.**
> >
> > **A5.** Since we assume that desired outcome examples from $G^+$ are given rather than its distribution, we address the combinational setting that requires assigning curriculum goals from the entire candidates in the replay buffer to the desired outcome examples..
> >
> > Firstly, we consider a bipartite graph $G({V_a, V_b}, E)$, where $V_a$ and $V_b$ are the set of the nodes representing states in the replay buffer, and desired goals respectively. Also, we set the weight of the edges as $E(s_a\in V_a, s_b\in V_b) = −w(s_a, s_b)$ (Eq. 11). Then, finding K edges with minimum cost $w$ connecting $V_a$ and $V_b$ can be solved by implementing Minimum Cost Maximum Flow algorithm, which is the well-studied problem in bipartite matching (Eq.11 and line 3 in Algorithm 1).
> >
> > - - -
> >
> > **Q6. A diagram/visual of different moving parts of the algorithm would be super helpful.**
> >
> > **A6.** Thank you for your helpful suggestions on the need for a diagram of OUTPACE. We added a diagram that visualizes an overview of our algorithm in the revised manuscript (due to the page limit, we added it in the Appendix). Also, we would like to attach an anonymous link to access the diagram for your convenience.
> >
> >
> > https://drive.google.com/file/d/151CkC4XTcxnY0EzroJaLYWhxadBm7VtE/view?usp=sharing
> >
> > - - -
> >
> > **Q7. More explanation and analysis of the results could be very useful for understanding the performance of the proposed method. In Figure 5, it seems a lot of baseline data are missing in many task domains but it is unclear whether there is no data or the success rates are zeros for all of them?**
> >
> >
> > **A7.** Thanks to your valuable comment, we recognized that it may appear that some baselines are missing in Figure 5. We note that **the curves are not visible as they overlap at zero success rate**. Following your suggestion, we added the following sentence in the caption of Figure 5.
> >
> > *Note that the curves of the baselines are not visible as they overlap at zero success rates in some environments.*
> >
> > If your concerns are still not addressed by the above explanation about missing baselines, we politely ask you to refer to our answer for another reviewer's similar question (the answer of Q1 for the reviewer KAqi). If this response is not sufficient enough to address your comments, please do not hesitate to let us know.
> >
> > - - -
> >
> > **Q8. The presentation of this work in general needs improvement. The language used is not formal and at the same time not easy to understand with grammatical errors.**
> >
> > **A8.** We tried to rewrite some phrases with formal language (e.g. : 'get an estimate of ...' &rarr; 'obtain an estimate of ...') and fix some grammatical errors (e.g. : 'datapoints' &rarr; 'data points') as much as possible. If there are still some weird phrases that hinder the reader's understanding, please let us know. If you could specify some examples, it would be really helpful for us to revise our work with better readability.
> >
> > - - -
> >
> > Thank you again for your valuable and insightful review.
> >
> > Please let us know if our responses have addressed your comments. If anything needs further clarification, please do not hesitate to let us know.
> > - - -

---

### Author Response · Authors · 2022-11-25
**Typo in the revised manuscript**

There is a typo in the revised manuscript.

On the 4th line after Eq. 4 (page 4)

"will predict the label e_n as rat class" -> "will predict the label e_n as its own class"

---

### Decision · Program_Chairs · 2023-01-20

**Decision:**

Accept: notable-top-25%

**Justification For Why Not Higher Score:**

In my view the authors could have evaluated on much richer and complex domains.

**Justification For Why Not Lower Score:**

Important contribution on automated curriculum learning with unanimous positive assessment by the reviewers.

**Metareview: Summary, Strengths And Weaknesses:**

I thank the authors for their submission and active engagement during the discussion period. The reviewers unanimously agree that this paper is worthy of publication. In particular, they found the work well motivated, novel, and with strong empirical results. In addition, the authors provide insightful ablations.

**Note From Pc:**

if the above contains the word "oral" or "spotlight" please see: "oral" presentation means -> notable-top-5% and "spotlight" means -> notable-top-25%. As stated in our emails, we are disassociating presentation type from AC recommendations